

# Hysteresis of the Greenland ice sheet from the Last Glacial Maximum to the future

Lucía Gutiérrez-González[1,2], Alexander Robinson[3], Jorge Alvarez-Solas[2,*], Ilaria Tabone[4], Jan Swierczek-Jereczek[1,2], Daniel Moreno-Parada[5], and Marisa Montoya[1,2,*]

[1]Department of Earth Physics and Astrophysics, Complutense University of Madrid, Madrid, Spain.
[2]Geosciences Institute, CSIC-UCM, Madrid, Spain.
[3]Alfred Wegener Institute, Helmholtz Centre for Polar and Marine Research, Potsdam, Germany.
[4]Department of Geophysics, University of Concepción, Concepción, Chile.
[5]Laboratoire de Glaciologie, Faculty of Sciences, Université Libre de Bruxelles, Bruxelles, Belgium

**Correspondence:** Lucía Gutiérrez-González (lucgut03@ucm.es) and Jorge Alvarez-Solas (alvarez.solas@igeo.ucm-csic.es)

**Abstract.** The Greenland Ice Sheet (GrIS) has undergone accelerated ice-mass loss in recent decades and it is expected to be one of the main contributors to global sea-level rise in the coming century. Due to the existence of positive feedbacks governing its mass balance, it is thought to be a tipping element of the Earth system. Its stability has been studied under temperatures ranging from the present day to a global warming of +4 K, showing a threshold behavior leading to an ice-free state for warmer

temperatures. However, its stability at lower temperatures has not been studied yet. Here we use the ice-sheet model Yelmo to obtain the stability diagram of the GrIS for the full range of glacial-interglacial temperatures, with regional summer air temperature anomalies relative to present extending from a climate representative of the Last Glacial Maximum (-12 K) to a warmer climate (+4 K). We find that the hysteresis persists in almost the entire studied range. Consistent with previous studies, a critical threshold is found between +1.2 and +1.8 K of regional summer air temperature anomaly, associated with atmospheric

feedbacks that are represented by the coupled regional energy balance model REMBO. In addition, a second threshold is found between -10 K and -9 K, that is mainly driven by ocean warming which triggers the marine ice-sheet instability in the glacial GrIS. The existence of this threshold is consistent with transient studies of the GrIS over the last glacial cycle.

## 1 Introduction

The Greenland ice sheet (GrIS) has undergone an accelerated ice-mass loss in the last decades, partly as a result of global

warming (Meredith et al., 2019; Hanna et al., 2021; Otosaka et al., 2023). There is a close agreement between the different methods used to estimate its mass loss, yielding a contribution of 14±2 mm to sea-level rise between 1992 and 2020 (Otosaka et al., 2023). As summarized in the Fifth (Bindoff et al., 2014) and Sixth (Eyring et al., 2021) Assessment Reports of the Intergovernmental Panel for Climate Change (IPCC AR5 and AR6, respectively), several studies attribute both the temperature increase and the sea-level rise to anthropogenic forcing (Bamber et al., 2019; Fox-Kemper et al., 2021; Marcos and Amores,

2014; Meredith et al., 2019; Slangen et al., 2014).





Ice-mass loss, once triggered, is amplified by at least two positive feedbacks: the melt-elevation and the melt-albedo feedback. As a consequence, the GrIS responds nonlinearly to an increase of temperature and shows a threshold behavior: if a temperature threshold is exceeded, the GrIS can reach a qualitatively different equilibrium state, with strongly reduced ice volume. The GrIS is therefore considered to be a tipping element (Lenton et al., 2008). Ice-sheet modelling studies furthermore
suggest that the GrIS shows multistability and hysteresis with respect to the temperature forcing (Robinson et al., 2012; Höning et al., 2023). As a result, if the rise in temperatures caused by anthropogenic climate change continues in the future, the GrIS could reach a new, ice-free equilibrium state, and it may not be able to recover its current extent and volume unless the temperature is decreased well below pre-industrial values. Therefore, the ice volume reduction could be irreversible on very long timescales. However, this does not mean that the changes would necessarily happen abruptly in time: due to the slow response
of the ice sheets to atmospheric changes, variations in volume take relatively long times to manifest (Alley et al., 2005). The ice-free equilibrium state works as an attractor to which the GrIS approaches in global warming scenarios. The final state of the ice sheet depends on the rate of global warming and the time above the critical threshold (Bochow et al., 2023).

Despite the widespread agreement concerning the threshold behavior of the GrIS, the precise values of the critical threshold for the future remain highly uncertain (Fox-Kemper et al., 2021). Simulations indicate a critical mean annual global temperature
threshold ranging between 1.6 K (Robinson et al., 2012) and 3.1 K (Gregory and Huybrechts, 2006) above pre-industrial values. In addition, the hysteresis behavior of the GrIS has only been studied under temperatures above present-day values (Robinson et al., 2012; Bochow et al., 2023; Höning et al., 2023). Ice-sheet volume hysteresis for lower temperatures has only been studied in the context of the complete Northern Hemisphere and as a function of the insolation (Calov and Ganopolski, 2005; Abe-Ouchi et al., 2013).

Here, our main goal is to investigate the GrIS critical thresholds and stability across a range of temperatures, from conditions representative of the Last Glacial Maximum (LGM; ca. 21 kyr), when regional summer atmospheric temperatures were approximately 12 K below present-day values (Kindler et al., 2014), to future globally warmer conditions, with a summer regional anomaly of +4K. To this end, we calibrate the ice-sheet model in order to correctly represent the GrIS both at the LGM and the present day. We investigate whether the hysteresis persists in colder climates and whether additional tipping points exist.
Finally, we characterize the change in equilibrium states at each transition, and we study the dominant mechanisms along the stability diagram.

The work is structured as follows: in Section 2 we describe the the ice-sheet model Yelmo coupled to the regional climate model REMBO used for our simulations and the experimental setup; in Section 3 we present and analyze the results of the experiments; in Section 4 we discuss the main results and relate them to the existing literature. Finally, the main conclusions
of the work are given in Section 5.





## 2 Methods

### 2.1 Model

We use the ice-sheet model Yelmo (Robinson et al., 2020) coupled with the regional energy-moisture balance atmospheric model REMBO (Robinson et al., 2010). Yelmo is a three-dimensional thermomechanical ice-sheet model that has been pre-
viously used to simulate the evolution of the Greenland (Bochow et al., 2023; Tabone et al., 2024), the North American (Moreno-Parada et al., 2023) and the Antarctic ice sheets (Blasco et al., 2021; Juarez-Martinez et al., 2024). To compute the ice velocities, Yelmo uses the depth-integrated-viscosity approximation (DIVA) that considers longitudinal, lateral, and vertical shear stresses (Goldberg, 2011). In DIVA, the effective viscosity and the horizontal velocity gradients are replaced by their depth-integrated values in the effective strain rate equation. This allows for a more efficient two-dimensional solution, which
yields close results to the computationally much more expensive Stokes problem (Robinson et al., 2022). A more detailed description of Yelmo is found in Robinson et al. (2020).

The basal friction is calculated using the regularized Coulomb friction law (Joughin et al., 2019) as follows:

$$\boldsymbol{\tau}_{\mathrm{b}} = c_{\mathrm{b}} \left( \frac{|\mathbf{u}_{\mathrm{b}}|}{|\mathbf{u}_{\mathrm{b}}| + u_0} \right)^q \frac{\mathbf{u}_{\mathrm{b}}}{|\mathbf{u}_{\mathrm{b}}|}, \tag{1}$$

where $\mathbf{u_b}$ is the basal velocity, $q$ the friction law exponent, $u_0$ the regularization term for the Coulomb law set at 100
$\mathrm{m \cdot yr^{-1}}$, and $c_b$ the basal yield stress, defined as:

$$c_{\mathrm{b}} = \tan\left(\phi\right) N. \tag{2}$$

Here, $N$ is the effective pressure, calculated following Bueler and van Pelt (2015). The till friction angle, $\phi$, is a linear function of bedrock elevation, with a lower limit of $1°$ when the bedrock is $700\,\mathrm{m}$ below sea level and an upper limit of $40°$ when the bedrock is $700\,\mathrm{m}$ above sea level. As in (Tabone et al., 2024), we impose a scaling factor of 0.1 on $c_b$ in the
northeastern basin. This accounts for the softer properties of the bed and allows for a better representation of the North East Greenland Ice Stream (NEGIS).

To represent glacial isostatic adjustment (GIA), Yelmo is coupled to the regional isostasy model FastIsostasy (Swierczek-Jereczek et al., 2024). We employ here an elastic lithosphere and a viscous mantle (ELVA) with a spatially homogeneous upper-mantle viscosity of $1 \cdot 10^{-21}\mathrm{Pa \cdot s}$ in the first $88\,\mathrm{km}$ and $2 \cdot 10^{-21}\mathrm{Pa \cdot s}$ in the following $600\,\mathrm{km}$, and gravitational interactions
between the ice sheet, the solid Earth and the ocean.

The surface mass balance (SMB) is calculated by the regional energy-moisture balance (REMBO) model, which follows an ITM scheme, described below in section 2.2. The geothermal heat flow is taken from (Shapiro and Ritzwoller, 2004) and is imposed as a boundary condition at a depth of $2\,\mathrm{km}$ into the bedrock. Submarine melting at the base of ice shelves is parametrized by a linear equation dependent on ocean temperature (Section 2.3). Calving follows the Von Mises stress criterion
(Morlighem et al., 2016) and it is defined as a function of the effective stress at the ice front (Lipscomb et al., 2019).



## 2.2 Atmospheric forcing

REMBO is a 2D regional climate model that simulates daily precipitation and temperature fields for Greenland, as well as the surface mass balance (SMB) through a one-layer snowpack model. The temperature and humidity over the ocean around Greenland are imposed as boundary conditions, for which the climatological mean of ERA-40 reanalysis (Uppala et al., 2005)
is used for the present day. The atmosphere over Greenland is then simulated through the vertically integrated energy balance and moisture balance equations. Additionally, REMBO takes into account the orography of Greenland and the planetary albedo, including its seasonal changes, thus reproducing the associated elevation and albedo feedbacks (Robinson et al., 2010). REMBO does not simulate the atmosphere's broader general circulation and has its own biases, particularly in precipitation. Although its resolution is low (100 km), the surface boundary conditions — and therefore the melt–elevation and melt–albedo
feedbacks — are computed at the ice-sheet model resolution of 16 km. As a result, the model is able to represent the key atmosphere–ice feedbacks relevant to this study at a relatively high resolution. It has been previously used to simulate both the past and the future evolution of the GrIS (Robinson et al., 2012, 2011; Calov et al., 2015; Bochow et al., 2023).

The SMB is defined as the difference between the accumulation and the runoff. REMBO follows the insolation-temperature melt (ITM) method, which was originally developed by (Pellicciotti et al., 2005) and (van den Berg et al., 2008). The surface
melt rate $M_s$ follows the linear equation:

$$M_s = \frac{1}{\rho_w L_m}[c + \lambda T + \tau_a(1 - \alpha_s)S], \tag{3}$$

where $\rho_w$ is the water density; $L_m$ is the latent heat of ice melting; $c$ and $\lambda$ are empirical parameters set to -55 $\mathrm{W \cdot m^{-2}}$ and -10 $\mathrm{W \cdot m^{-2} K^{-1}}$; $\tau_a$ is the transmissivity of the atmosphere; $\alpha_s$ the surface albedo; and $S$ the insolation at the top of the atmosphere. Therefore, the dependence of melting on both absorbed insolation and temperature is taken into account. The
surface albedo is parametrized as a function of the snow thickness and the type of ground surface (Robinson et al., 2010).

REMBO is bidirectionally coupled to the ice-sheet model. The SMB and surface temperature obtained by REMBO force the ice-sheet evolution in Yelmo and the evolution of the topography calculated in Yelmo is an input for REMBO (Robinson et al., 2010).

To reduce the model bias, we applied a simple direct correction method (Wetterhall et al., 2012) to the precipitation field
based on $P_{MAR}$, the 1961–1990 mean of the monthly precipitation from Tedesco et al. (2023). Thus, the corrected precipitation is:

$$P_{corr} = P \cdot \delta P, \quad \text{where} \quad \delta P = \frac{P_{MAR}}{P_{REMBO,ref}} \tag{4}$$

where $P$ is the precipitation without correction, and $P_{REMBO,ref}$ is the precipitation simulated by REMBO for the present day without correction. In order to maintain a consistent field, $\delta P$ is restricted to values between 0.6 and 1.4. Furthermore, we
apply Gaussian smoothing to the reference precipitation $P_{MAR}$ so that we only correct large-scale features. This correction serves to improve the agreement of the seasonality and overall magnitude of precipitation, particularly in the north.

To force REMBO, we impose monthly temperature anomalies at the boundaries. Here, as in Robinson et al. (2012), we assume that annual temperatures follow a sinusoidal cycle where $\Delta T_{DJF} = 2\Delta T_{JJA}$, so that the annual temperature anomaly



is $\Delta T_{ann} = (\Delta T_{DJF} + \Delta T_{JJA})/2 = 1.5\Delta T_{JJA}$. Since temperature during the summer months is more important for melting,
we use $\Delta T_{JJA}$ as input.

## 2.3 Oceanic forcing

The ocean forcing of the GrIS is treated in very different manners throughout modelling studies. Some fully neglect it, taking
into account only atmospheric forcings (Stone et al., 2013) while others parameterize sub-shelf melting as a function of the
ice-shelf draft (i.e., the ice-thickness below the ocean surface) (Bradley et al., 2018). Still, others do not use an explicit forcing
but constrain their model with relative sea level (Simpson et al., 2009; Lecavalier et al., 2014). Finally, some include parameter-
izations of basal melting as a function of ocean temperatures (e.g. Tabone et al., 2018). Results show that ice-ocean interactions
play a key role in the past GrIS evolution, notably at times and in areas where the ocean-ice contact zone is largest, such as
the LGM, when the GrIS margins reached the continental shelf break. Even if present marine-terminating glaciers cover only
a small part of Greenland in the present day, their advance and retreat significantly affects the ice geometry and dynamics in
the continental zone, and this effect is amplified as the ice-ocean contact zone expands (Tabone et al., 2018).

As in (Tabone et al., 2018), we formulate the basal melting rate at the grounding line $B_{gl}$ using an anomaly approach:

$$B_{gl}(t) = \kappa\Delta T_{ocn}(t) + B_{ref}, \tag{5}$$

where $\kappa = 15\,\mathrm{m\cdot yr^{-1}\cdot {}^{\circ}C^{-1}}$ is the heat flux between the ice and the ocean at the ice-ocean interface; $\Delta T_{ocn}(t)$ is the anomaly
between the ocean temperature at time $t$ and at present day; and $B_{ref}$ represents the basal melting rate at the present day. Here,
$B_{ref}$ is set at $50\,\mathrm{m\cdot yr^{-1}}$ following observations of basal melting at the grounding line (Wilson et al., 2017). All parameters
in Eq. 5 are assumed to be spatially uniform along the marine boundaries of the GrIS. Using a single value is a coarse ap-
proximation; however, in the absence of a complete sub-shelf melt map for Greenland, constructing a reliable 2D spatial field
would be subjected to large uncertainties. Therefore, for the idealized experiments conducted in this study, we consider this
approximation to be appropriate. $B_{gl}$ is limited to positive values, which implies that no refreezing is allowed at the grounding
line.

$\Delta T_{ocn}$ is related to the annual atmospheric temperature anomaly $\Delta T_{ann}$. This relationship is parameterized linearly (Golledge
et al., 2015) as follows:

$$\Delta T_{ocn} = 0.25\Delta T_{ann}. \tag{6}$$

On the other hand, for purely floating ice shelves the sub-shelf basal melting rate $B_{sh}$ is lower than that at the grounding
line:

$$B_{sh} = \gamma B_{gl}, \tag{7}$$

where $\gamma = 0.1$, in accordance with observations of Greenland glaciers (Wilson et al., 2017).





## 2.4 Experimental design

To study the stability of the GrIS, warming (retreat) and cooling (regrowth) simulations were performed, starting at different
initial states, respectively: an LGM-like state and a virtually ice-free state (Fig. 1). To obtain these initial states, we performed
two spin-up experiments (one for each branch), initialized with present-day topography and ice thickness (Morlighem et al.,
2017, 2022), and ran them for 60 kyr with a constant forcing. The ice-sheet model has 10 vertical layers in sigma-coordinates
and a horizontal resolution of 16 km, the same as in other recent studies examining the stability of the GrIS (Bochow et al.,
2023; Höning et al., 2023). While a 16 km resolution may misrepresent certain processes, it enables long-term simulations that
otherwise would be not possible due to the computational cost (see Sec. A). At this horizontal resolution, interaction with the
ocean is not resolved in the narrowest fjords. For this reason, as in (Tabone et al., 2024), the bedrock elevation is reduced by 1
$-1.5\sigma$, where $\sigma$ represents its standard deviation, in the areas in contact with the ocean and higher uncertainty in the value of
the bedrock elevation.

The LGM-like (hereafter LGM) state (Fig. 1a) was obtained by imposing a regional summer air temperature anomaly of
-12 K (Kindler et al., 2014) and an ocean temperature anomaly of -4.5 K, according to Eq. 6. This is not an exact reconstruction
of the LGM but rather an LGM-like state, as it retains present-day sea level, insolation, and $CO_2$ conditions. This approach
allows us to isolate the effect of temperature on ice-sheet stability without additional forcing changes. The virtually ice-free
state was obtained through the simulation of a warmer state (Fig. 1b), with a summer air temperature anomaly of +4 K and an
ocean temperature anomaly of +1.5 K.

For the LGM state (Fig. 1a), we obtain an ice sheet that is considerably larger than the present one. It has a simulated exten-
sion of 3.07 million $km^2$, a total volume of 5.15 million $km^3$ and a volume above flotation of 4.6 million $km^3$ corresponding to
an anomaly of 3.66 m of sea level equivalent (SLE) for the LGM GrIS. The latter value is within the range reported in literature,
ranging between 2.6 and 4.7 m SLE (Buizert et al., 2018; Lecavalier et al., 2014; Tabone et al., 2018; Bradley et al., 2018).
Low atmospheric and ocean temperatures allow for the expansion of the GrIS up to the continental shelf break, with floating
ice shelves appearing in the southeast and in Baffin Bay, fed by multiple ice streams. Large agreement is found between the
simulated and the reconstructed margins for the LGM (Lecavalier et al., 2014). An exception is the northeastern area, where
our simulation reaches the continental shelf break while the reconstruction by Lecavalier et al. (2014) does not. Whether the
GrIS reached the shelf break in that region or not is still under debate (Evans et al., 2009; Arndt et al., 2017; Leger et al.,
2024) but a more recent study based on high-resolution hydro-acoustic data suggests that was the case (Arndt et al., 2017), thus
matching our simulations. What is almost certain is that the GrIS expanded beyond the limits suggested by Lecavalier et al.
(2014), whose reconstruction for the LGM appears to be a lower limit as compared to the former studies.

With a regional summer air temperature anomaly of +4 K the ice sheet is drastically reduced both in volume and extent
(Fig. 1b). Ice persists only in the high-mountain areas: at the southern tip of Greenland and in the easternmost region, around
the Watkins range. Its mass balance is almost zero in the ice-sheet interior (all incoming accumulation is melted away) and
negative at its margins, especially in areas in contact with the ocean (not shown).



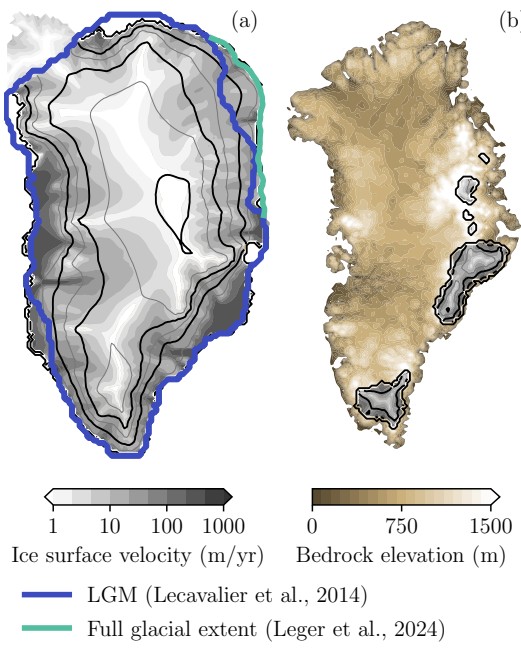

**Figure 1.** Final state of the spin-up simulation for a) the LGM-like state (initial state of the warming branch), obtained through a regional summer air temperature anomaly of -12 K, and b) a virtually ice-free state of +4 K (initial state of the cooling branch). The black contour lines indicate the surface elevation every 500 m starting from 0 and with a thicker line every 1000 m. In a) the blue line indicates the reconstructed ice-sheet margin of Lecavalier et al. (2014); the green line is the maximum extent of grounded ice from the full glacial extent (18-16 kyr BP) in the northeast region by Leger et al. (2024). Note that the lightly shaded area in the northwest indicates the part of the simulated ice sheet that is not taken into account for the volume calculation.

Finally, we performed a third spin-up for present-day conditions, for which we find a close agreement between our simulation and the observations (see Fig. A1 in the appendix).

To study the wider stability of the GrIS, we perform a series of simulations, outlined in Fig. 2. First, quasi-equilibrium simulations were performed with different rates of forcing (between $5 \cdot 10^{-3}\,\mathrm{K} \cdot \mathrm{yr}^{-1}$ and $1 \cdot 10^{-5}\,\mathrm{K} \cdot \mathrm{yr}^{-1}$) in a range of regional

summer air temperature anomalies from -12 K to +4 K relative to the average value between 1958–2001, which is used for the boundary conditions of the model as in (Robinson et al., 2012; Bochow et al., 2023). For the warming experiments, a positive constant temperature anomaly rate is applied to the LGM state, until a temperature anomaly of +4 K, relative to the average value, is reached. In parallel, for the cooling experiments a negative constant temperature anomaly rate is applied, starting from the virtually ice-free state until a temperature anomaly of -12 K is reached. Therefore, to achieve the full temperature anomaly

range of 16 K from the initial state, the durations of the simulations vary between 3.200 years and 1.6 million years, depending on the forcing rate.





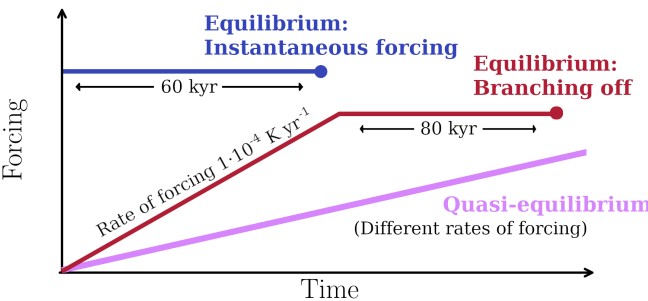

**Figure 2.** Diagram of the forcing in the different simulations performed to obtain the stability diagram for the warming branch. The quasi-equilibrium simulations consist of ramping-up at different rates of forcing. In contrast, each equilibrium state is the final state (represented by a triangle) of one simulation with (1) instantaneous forcing and (2) branching off from quasi-equilibrium simulations with a rate of forcing of $1 \cdot 10^{-4} \mathrm{K} \cdot \mathrm{yr}^{-1}$. For the cooling branch simulations, the forcing is the same but with negative trends. The initial state of the three experiments is obtained from a steady-state spin-up simulation of the LGM (Fig. 1a) for the warming branch and a steady-state spin-up simulation under +4 K warming for the cooling branch (Fig. 1b).

In addition, equilibrium states are derived through two types of simulations: (1) applying an instantaneous summer air-temperature anomaly to the initial state during 60 kyr across different temperature levels ranging from -12 K to +4 K, in 1 K increments; and (2) similar to Garbe et al. (2020), branching off from the $1 \cdot 10^{-4} \mathrm{K} \cdot \mathrm{yr}^{-1}$ quasi-equilibrium simulation, then maintaining a constant temperature at different forcing levels (ranging from -12 K to +4 K for the atmospheric temperature, with 1 K increments and 0.2 K increments near the tipping points) for 80 kyr, 120 kyr or 400 kyr, depending on the time required for the simulations to stabilize.

By decreasing the forcing rates in the quasi-equilibrium simulation, we bring the transient simulations closer to the equilibrium states. Ideally, this would be achieved with an infinitely slow forcing rate, which is however impossible in practical terms. Nonetheless, the combination of the transient quasi-equilibrium simulations with the equilibrium states allows us to perform an exhaustive analysis that characterizes the stability phase space over the chosen range of study— where the tipping point values are taken from the branching-off experiments, as these are the most reliable due to their gradual convergence toward equilibrium.

## 3 Results

### 3.1 Bifurcation diagram

Through the different equilibrium and quasi-equilibrium experiments of the GrIS, we obtain two different branches (warming and cooling, or retreat and regrowth) that constitute the bifurcation diagram (Fig. 3a). As the forcing rates decrease (and therefore the forcing periods increase) the transient quasi-equilibrium response converges to the equilibrium states. This is a





**Figure 3.** a) Stability diagram of the GrIS: volume above flotation versus regional summer air temperature anomaly. The solid lines indicate the quasi-equilibrium simulations at different rates of forcing. The triangles indicate the final states of the instantaneous-forcing simulations (gray) and of the branching-off-from-quasi-equilibrium simulations (black); the direction of the triangles indicates the sense of the forcing (to the right in the warming or retreat branch and to the left in the cooling or regrowth branch). The light red shaded area above the warming branch represents the projection of the basin of attraction for the warming branch, i.e., the points for which the mass balance is negative under the given forcing and the ice sheet is expected to retreat until reaching the equilibrium state. The light blue shading indicates the analogous region for the cooling branch, identifying the points where the ice sheet is expected to grow. b)-f) GrIS configuration for the equilibrium states marked in a). The blue thick lines in b) and c) indicate the ice-sheet margin of (Lecavalier et al., 2014) and the green line is the maximum extent of grounded ice from the full glacial extent (18-16 kyr BP) in the northeast region by Leger et al. (2024). The red thick lines in d)-f) indicate the present-day margin of Morlighem et al. (2017). The black contour lines indicate the surface elevation every 500 m starting from 0 and with a thicker line every 1000 m.



consequence of the large inertia of the ice sheet: only long forcing periods allow for the ice-sheet response to the forcing to fully develop and to approach the equilibrium states. In the slowest simulation ($1 \cdot 10^{-5} \mathrm{K} \cdot \mathrm{yr}^{-1}$), which takes 1.6 million years, the volumes coincide with the equilibrium values for the entire interval, except in the vicinity of the tipping points, where the ice sheet takes even longer to stabilize.

We identify two bifurcation points in the warming branch. As noted earlier, their values are best determined from the branching-off equilibrium simulation, which provides the most reliable representation of the equilibrium states. The first one, between -10 K and -9 K, begins with the loss of most of the GrIS ice shelves, followed by the retreat of the northeast margin (the state of the ice sheet before and after the transition is shown in Fig. 3b and Fig. 3c and is analyzed in greater depth in Section 3.3). Once a temperature anomaly of -9 K is reached, a gradual decrease in volume begins, showing an almost linear trend with a slope close to zero relative to the forcing. This behavior persists until a temperature anomaly of 1.2 K is reached (Fig. 3d), at which point the positive melt-albedo and melt-elevation (atmospheric) feedbacks are triggered (Sec. 3.2). This second tipping point is identified in this study between +1.2 K and +1.8 K.

Fig. 4 shows a zoomed-in view around this tipping point. Near the bifurcation point—particularly at +1.4 Kand +1.6 K—in both equilibrium branches of the branching-off experiments, the ice sheet oscillates between two states (shown in more detail in Fig. B1). These oscillations show periods ranging from 70 kyr to 120 kyr. At +1.8 K, the GrIS reaches a stable virtually ice-free state with a volume of 1.58 m of SLE (Fig. 3e). In this transition, we move from an ice sheet that almost completely covers the surface of Greenland to a virtually ice-free state, with some residual ice only at the highest altitudes, mainly in the Watkins Range and the southern tip.

In the regrowth (cooling) branch, the GrIS starts at its virtually ice-free state (Fig. 1b), with $\Delta T_{JJA}$=+4 K, and it is cooled. The ice sheet remains nearly ice-free until 1.6 K, where oscillations resume, as in the warming branch. The ice sheet remains in an intermediate state (Fig. 3f) until 0 K, with no ice in the north due to the high temperatures and the low precipitation in the area.

In this cooling branch, quasi-equilibrium simulations are farther away from equilibrium than in the warming branch for the same rates, especially during the critical transition (Fig. 3a, quasi-equilibrium simulations with rates of forcing between $10^{-5}$ – $10^{-3} \mathrm{K} \cdot \mathrm{yr}^{-1}$ and the temperature range 0 K – +1.8 K). Starting from the ice-free state, the recovery of the ice sheet requires lower temperatures and more time. This is due to the fact that melting or calving can occur almost instantaneously with high rates, while the processes of accumulation and regrowth are slower and can require thousands of years. Between 0 K and -9 K the ice sheet does not show large variations, but just a linear regrowth with small sensitivity to temperature changes. Finally, from -9 K onwards, the sensitivity is increased again: the ice shelves are recovered, and the northeastern margin expands until it reaches the continental shelf break. By -12 K, it attains a volume above flotation of 4.45 million km$^3$, almost converging to the initial state of the warming branch.

Hysteresis persists throughout almost the entire temperature range (between -10 K and +1.4 K), with the largest volume difference for temperature anomalies between 0 and +1.4 K. It is important to highlight how, for the branching-off equilibrium states, the hysteresis is narrower and the regrowth in the cooling branch allows intermediate equilibrium states like that shown in Fig. 3f. At $\Delta T_{JJA}$=+1.2 K multistability becomes evident, with two different ice-sheet configurations (Fig. 3 d and f) and





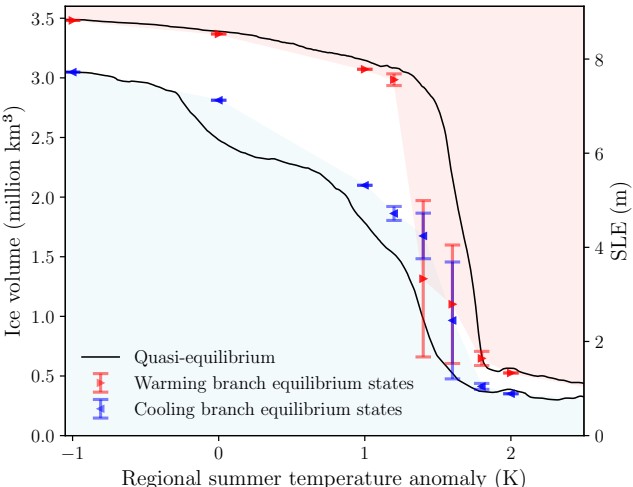

**Figure 4.** Enlarged section of the stability diagram shown in Fig. 3, focusing on the temperature range from -1 to 2.5 K. Solid lines indicate the slowest quasi-equilibrium simulation ($1 \cdot 10^{-5}$ K·yr$^{-1}$). Red (blue) triangles represent the branching-off equilibrium states of the warming (cooling) branch. In cases with regional summer temperature anomalies of 1.2 K, 1.4 K, 1.6 K, and 1.8 K, the simulations show stationary oscillations and stabilize in a dynamic state. In these cases, unlike in Fig. 3, the triangles represent mean values, and the whiskers indicate the amplitude of the oscillation.

respective volumes of 2.99 and 1.82 million km$^3$. These results clearly show the impact of atmospheric feedbacks related to

elevation and albedo, as well as the irreversible nature of this transition on long time scales: if the present-day GrIS collapses, even if temperatures return to present-day values, there would still be a 1.4 m SLE difference.

Within the range from -9 K to 0 K, the differences between branches especially affect the ice thickness in the margins. For example, for a summer air temperature anomaly of -5 K (Fig. 5a-c), the equilibrium ice sheet on the warming branch extends well beyond the margin of the equilibrium state in the cooling branch for the same temperature and has a 12% larger

volume. This nonlinearity originates from different mechanisms. To explain these, we analyze the ice-sheet evolution at selected transects (1 and 2; Fig. 5d,e). These are located in areas with the greatest differences between the grounding lines, following the ice-flow streamlines. Along these transects, the Disko Island (Qeqertarsuaq in Greenlandic) and a bathymetric peak in the northeast act as pinning points for the warming branch equilibrium states (analogous to those found for the the Antarctic Ice Sheet in (Favier et al., 2016, e.g.,)). These slow down the outflow and allow the ice sheet to have a higher volume.

On the other hand, in the cooling branch the ice sheet shows a more irregular margin than in the warming branch. As it grows from the interior, it encounters the coastline (which is highly irregular) as a boundary, where ocean-induced melting occurs, preventing further growth. In contrast, in the warming branch, since it originates from an ice sheet in equilibrium and of greater extent (the LGM-like state), the continuity of the ice allows basal melting caused by the ocean to melt the margin more uniformly, resulting in a more compact and regular ice sheet shape. This leads to a larger perimeter in the cooling branch:





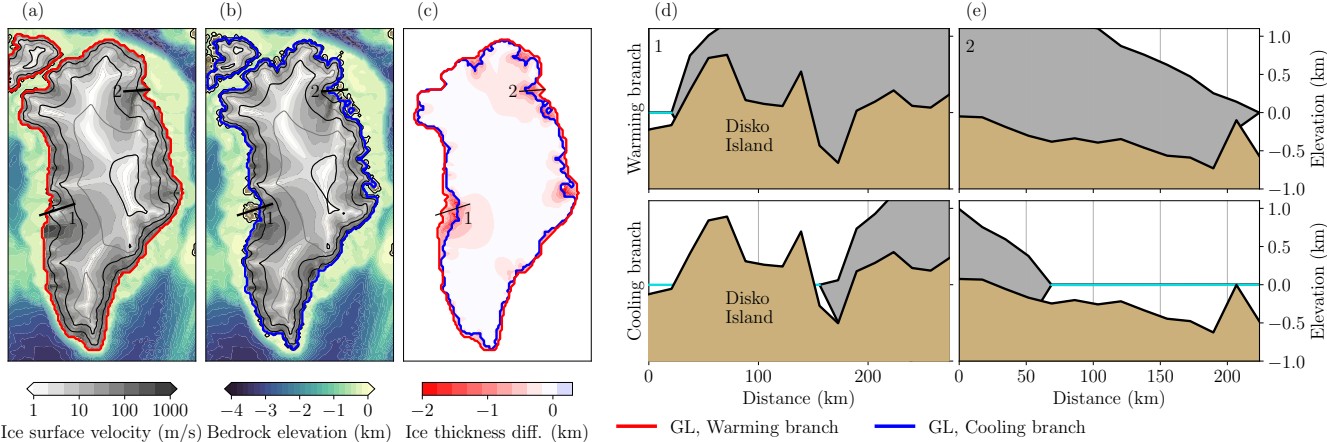

**Figure 5.** Differences between the two equilibrium states at $\Delta T_{JJA}$=-5 K (from the branching-off experiments). a) and b) show the surface velocities and the grounding line for the warming (red line) and the cooling (blue line) branch, respectively. The black contour lines indicate the surface elevation every 500 m starting from 0 and with a thicker line every 1000 m. c) Ice thickness difference between the two branches (cooling branch minus warming branch). d) and e) show the bedrock elevation (brown), the ice thickness (gray) and the sea level (blue line) for both branches at transects 1 and 2 indicated in a)-c).

the cooling branch state has a margin of 40.448 km, whereas the warming branch state's margin is 26.496 km long. As a result, even though some areas in the cooling branch have not yet reached the coastline, the total margin in contact with the ocean is greater (9.984 km vs. 6.496 km in the warming branch). This generates a higher basal melting and calving, making the regrowth difficult and preventing the ice sheet from reaching the size of the warming branch at the same temperature.

## 3.2   Forcing mechanisms: the ocean and the atmosphere

In order to analyze the mechanisms leading to ice loss and recovery along the stability diagram, we show the surface, basal and lateral mass fluxes in Fig. 6. This analysis can only be performed with the quasi-equilibrium simulations, so we selected the simulation with the lowest forcing rate ($1 \cdot 10^{-5}$ K $\cdot$ yr$^{-1}$), as it is the closest to equilibrium. As we saw in the previous section, for each quasi-equilibrium simulation, the system is farthest from equilibrium near the tipping points (shaded gray). This is reflected in the total mass balance (Fig. 6), which remains stable over the entire bifurcation diagram except in the proximity of

the tipping points, where it becomes negative for the warming branch and positive for the cooling branch.

For the warming branch, the SMB increases between -12 K and -4 K as higher temperatures allow for more precipitation. However, from -4 K onwards, ablation increases and the SMB gradually decreases. Finally, at ∼+1.5 K elevation and albedo feedbacks are triggered and the SMB drops abruptly and becomes negative. The basal mass balance (BMB) is negligible between -12 K and -10.5 K due to the fact that low ocean temperatures do not allow for sub-shelf melting, and to the small

value of the melting under grounded ice. When temperatures rise enough to produce sub-shelf melting, an ice-shelf reduction,





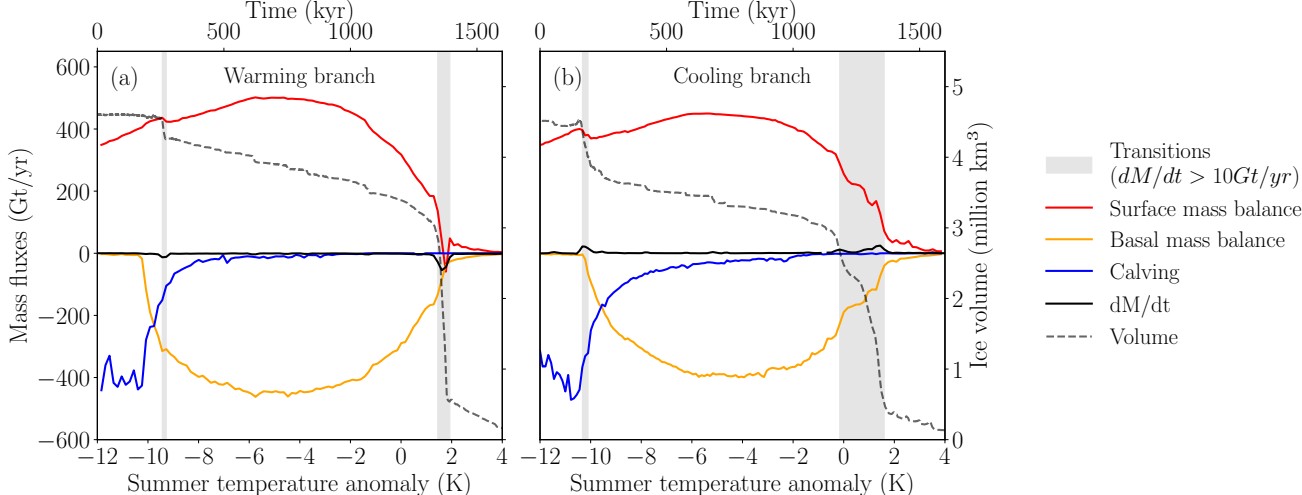

**Figure 6.** Mass balance contributions of the quasi-equilibrium simulation of $1 \cdot 10^{-5} \mathrm{K} \cdot \mathrm{yr}^{-1}$. a) warming; b) cooling branch. The gray shaded bars indicate the regions in which, despite the slow forcing rates, the system deviates from equilibrium due to the proximity to the tipping points. The black dashed line represents the total volume, with its axis located on the right-hand side.

a margin retreat, and a decrease in the ice-ocean contact zones is induced. This eventually causes a reduction in calving. BMB continues the increase with ocean warming until around -5K. After that point, sub-shelf melting begins to decrease as the contact with the ocean strongly diminishes, becoming almost negligible by +2K. Calving is highest in the temperature ranges where ice shelves are present (-12 K to -10.5 K). It decreases, as the BMB does, when the contact with the ocean decreases.

Therefore, before the first transition (at ∼–9.5 K for this quasi-equilibrium simulation), SMB is increasing, so only the sub-shelf basal melting —which intensifies with rising ocean temperatures— can drive the response, directly triggering ice-mass loss and accelerating ice dynamics. In contrast, during the second transition (at ∼+2 K), the influence of the ocean diminishes, as BMB decreases in amplitude and therefore cannot account for the observed mass loss. Instead, SMB decreases sharply, implying a positive feedback mechanism where surface forcing becomes the dominant driver.

In the cooling branch, surface processes are responsible for the ice-sheet growth, with accumulation increasing and ablation decreasing as temperature decreases. Basal melting and calving also increase as both the ice sheet and its contact zones with the ocean increase. Calving, in particular, strongly increases at around -9 K, when ice shelves recover and temperatures are too low to allow for basal melt. The transitions in this regrowth branch are more gradual than in the retreat branch, since the ice growth processes are slower.

## 3.3 Regional study

We next analyze the regional changes in ice-sheet volume by dividing the GrIS into three main regions: north, west, and southeast (Fig. 7). This division is based on grouping basins with similar topographical, atmospheric, and oceanic characteristics.





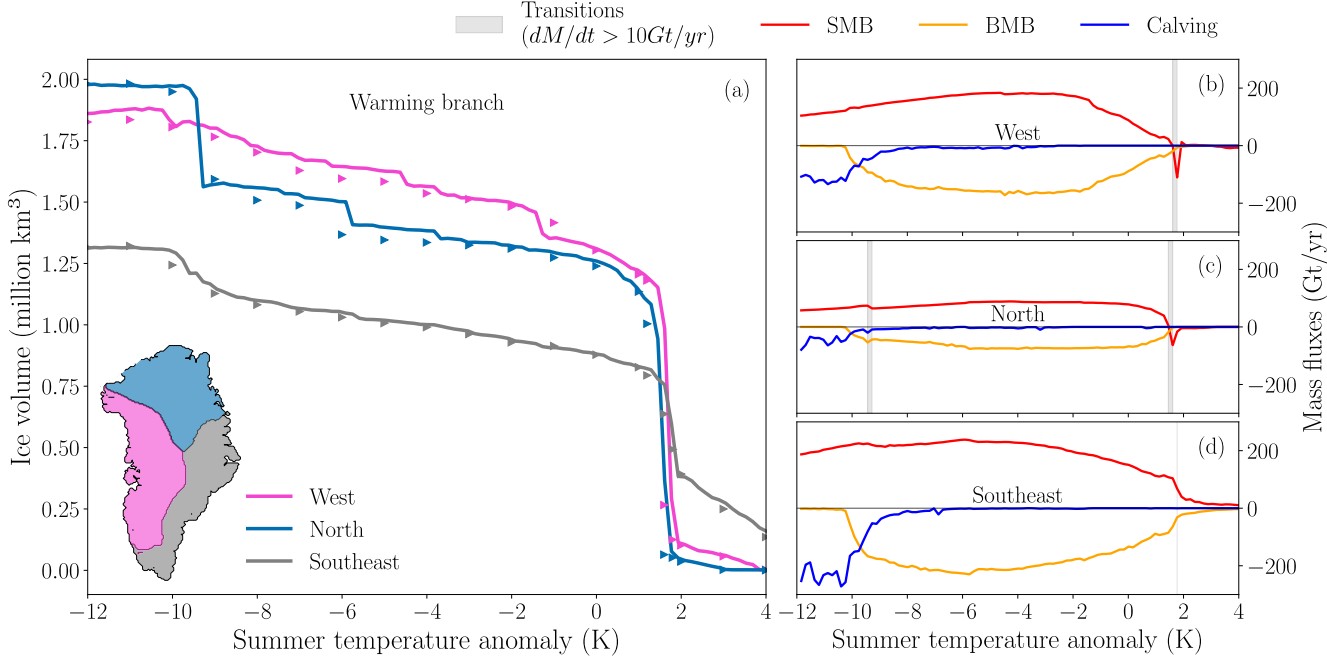

**Figure 7.** Warming branch of the stability diagram separated into three regions (defined in inset) adapted from Zwally et al. (2012)'s basins. a) Total volume (grounded and floating) of each region and b), c) and d) the corresponding mass fluxes for each region separately. Solid lines represent the quasi-equilibrium simulation with a rate of forcing of $1 \cdot 10^{-5} \, \mathrm{K} \cdot \mathrm{yr}^{-1}$, the triangles in a) show the equilibrium states from the branching-off experiments.

The northern region is formed by the basins at high latitudes around the Arctic Ocean, including also the North East Greenland Ice Stream (NEGIS) area (basins 1 and 2 in Zwally et al., 2012); the western region comprises basins discharging ice into Baffin Bay (basins 6, 7 and 8 in Zwally et al., 2012); and the southeastern region includes basins with mountain chains, where the ice persists for higher temperatures (basins 3, 4 and 5 in Zwally et al., 2012).

In Fig. 7, we can see that while the first tipping point has a regional character, affecting only the northern zone (the only one with $dM/dt > 10Gt/yr$), the second one is a global tipping point that influences the entire ice sheet. Additionally, when comparing the mass fluxes in the three regions in the quasi-equilibrium simulation with the lowest forcing rate as above (Figs. 7b-d), we find that in the northern area there is very little accumulation throughout the phase space. In contrast, the southeast area has a higher elevation and the SMB is significantly higher, with positive values even after exceeding the second tipping point.

Before reaching the first tipping point, the volume decreases in the west and southeast, where the ice shelves retreat as temperature rises and basal melting is activated. In contrast, in the northeast –where the ice is grounded and its thickness is higher– the volume remains nearly constant until the temperature anomaly reaches -9.4 K (in this quasi-equilibrium simulation), when the abrupt ice loss occurs. This rapid loss is driven by basal melting but also by the acceleration of ice flow in the area, a



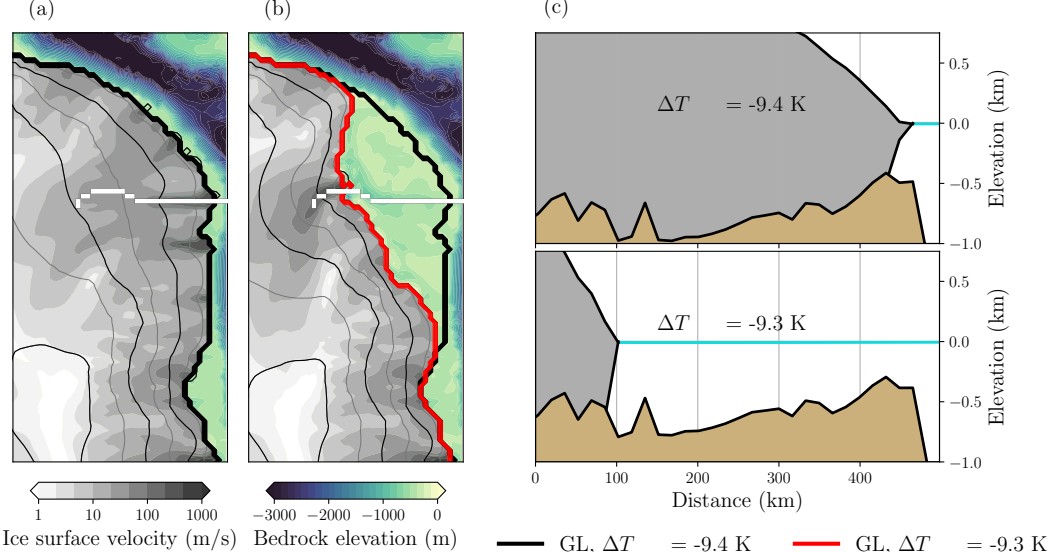

**Figure 8.** MISI in the northeast GrIS. a) and b) show the surface velocities before ($\Delta T_{JJA} = -9.4$ K) and after ($\Delta T_{JJA} = -9.3$ K) crossing the tipping point in the quasi-equilibrium warming branch simulation (rate of forcing of $1 \cdot 10^{-5}$ K $\cdot$ yr$^{-1}$). Black and red lines indicate the grounding line in both states and the black contours represent the surface elevation every 500 m and with a thicker line every 1000 m. c) shows the bedrock elevation (brown), the ice thickness (gray) and the sea level (blue line) for the transect 1 indicated in a) and b) with a white line.

consequence of the initial melting effect. This leads to a spike in calving of the grounded ice at -9.5 K (Fig. 7c) and the onset of margin losses. Before the tipping point (Fig. 8) the ice sheet remains grounded on a flat or prograde slope, but when the retreat starts, the ice-sheet margin goes through a mostly retrograde slope, where the grounding line is prone to be unstable due

to the marine ice-sheet instability (MISI) described by (Schoof, 2007) and studied in different areas of Greenland (Khan et al., 2020). This triggers a major retreat, and the losses are not compensated by precipitation, as accumulation is minimal at these latitudes. This is shown in Fig. 7c (north), where the SMB is lower than in the southeast and west (Figs. 7b and 7d), preventing the recovery of the ice sheet in this region.

At the second tipping point, the quasi-equilibrium simulation shows that losses also occur slightly earlier in the northern

and western regions, where the decline of the SMB is more notable than in the southeast (Figs. 7b to d). However, when the ice sheet reaches equilibrium for $\Delta T_{JJA}$=+1.8 K, the feedbacks that have triggered the ice losses result in the collapse of the entire ice sheet. Therefore, once ice loss begins in these regions, if the forcing persists in time, the complete loss of the GrIS becomes inevitable. Finally, as we saw in Fig. 1b, at 4 K the ice only remains in the southeast zone.



## 4 Discussion

We have mapped the stability diagram of the GrIS for the full range of temperatures it would experience, from the Last Glacial Maximum to future warming. We show the existence of hysteresis for almost the whole temperature range. This indicates that, once the GrIS retreats, cooling back to the original temperature is not sufficient to recover the original ice-sheet configuration, in agreement with previous studies (Robinson et al., 2012; Höning et al., 2023).

In our diagram, we identified two bifurcation points: one for colder climates and another one for warmer climates. These
tipping points divide the bifurcation diagram into three ranges, each with a distinct almost linear trend in ice-volume changes with respect to regional summer air temperatures. The transition from one regime to another depends on the rate and duration of the applied forcing once the threshold has been crossed. If the warming persists over time, the ice sheet will tend to the equilibrium state corresponding to that temperature anomaly. However, a rapid increase in temperature, if not sustained over time, does not lead to a total loss of ice. This allows for overshooting of the thresholds without losing the complete ice sheet as
shown by Bochow et al. (2023). For example, when $\Delta T_{JJA}$=+4 K is reached in the simulation with the fastest rate of forcing $(5\cdot10^{-3}\text{K}\cdot\text{yr}^{-1}$, pink line in Fig. 3a), the threshold is crossed by several degrees but the ice sheet still has a (transient) volume of 4.21 million km$^3$, even greater than the present-day value.

To our knowledge, the existence of a MISI-driven bifurcation between -10 K and -9 K has not been described in the literature to date. Reconstructions of the last deglaciation investigate the transition from the LGM to the present (Lecavalier et al., 2014;
Tabone et al., 2018; Leger et al., 2024). These are very different experiments from ours. Nevertheless, they provide useful information for comparative analysis and further support our findings. Lecavalier et al. (2014) show that the GrIS margin experienced a rapid general retreat at ∼16 to 10 kyr BP. It began gradually in the south, while in the north it was abrupt and occurred slightly later, with higher temperatures. These changes are also abrupt in the simulations of (Tabone et al., 2018), additionally highlighting the importance of oceanic forcing in the transitions between glacial and interglacial periods. (Winsor
et al., 2015) further emphasize the critical role of oceanic forcing as the main driver of retreat in the southwest margin, where marine-terminating glaciers are prevalent. Although these are transient simulations that deviate from equilibrium, they also show an abrupt transition largely influenced by ocean forcing above a certain temperature. In addition, as in our diagram, the southern region shows the greatest vulnerability, retreating at lower temperatures than the northern one. Moreover, although the retreat in the northern margin begins somewhat later, it occurs more rapidly. In addition, (Larsen et al., 2018) show that
the northeastern margin has experienced large fluctuations in the last 45 kyr, showing a large sensitivity to changes in ocean temperatures. In our study, the retreat in this area is driven by the ocean forcing, which accelerates the ice flux and triggers the MISI mechanism along the regions where the bedrock has a retrograde slope. In studies of the last deglaciation, instability is not necessary to explain the rapid retreat of the northeastern margin, as the temperature increase after the Younger Dryas was rapid and very large. However, through our equilibrium and quasi-equilibrium experiments, we find that nonlinear behavior
is indeed triggered by oceanic forcing. The exact value of this tipping point is heavily influenced by the model setup (see the Appendix C), as the activation of basal melting is determined by Equation 5. However, the parameters of this equation



correspond to observable physical data ($B_{ref}$) and produce a configuration capable of accurately representing the GrIS during both the LGM and the present-day, lending confidence to our results.

The second bifurcation point identified here is located between +1.2 and +1.8 K of regional summer air temperature anomaly.
Between these two states, the GrIS equilibrium states oscillate. Similar behaviour on long time scales was previously shown by Zeitz et al. (2022), and attributed to the interaction between the positive elevation feedback and the negative isostasy feedback. We do not analyze this effect and its consequences in detail, as it is beyond the scope of this work, but it appears to follow the mechanisms thoroughly characterized by Zeitz et al. (2022). In the vicinity of the bifurcation point, the ice sheet is unstable, and in all cases, once the +1.2 K threshold is exceeded, it undergoes a substantial retreat.

The hysteresis width found here, in the range of $0-2$ K, is somewhat narrower than in previous studies. Although this is not reflected in the quasi-equilibrium simulations, equilibrium states (black triangles in Fig. 3) display an ice regrowth that starts at higher temperatures than in previous studies. Between 0 and +1.4 K of regional summer air temperature anomaly, in the cooling branch, there are intermediate states for which the GrIS shows a partial recovery (Fig. 3f). Ice-sheet regrowth in other studies was found at +0.3 K (Robinson et al., 2012), or between -0.5 and +0.5 K depending on the level of previous warming (Höning
et al., 2023). We have not investigated in depth the reasons for these differences, but it is conceivable that they are due to structural differences between models. For instance, Robinson et al. (2012) employed the SICOPOLIS ice-sheet model, which relies on a different approximation of ice dynamics (SIA vs. DIVA), a different sliding law parametrization, and a different treatment of GIA, the elastic lithosphere and relaxed asthenosphere (ELRA) vs. ELVA. Additional differences include the representation of basal hydrology and the inherent numerical schemes. Moreover, our bedrock topography currently includes
much more detail than the older studies, which can also influence the ice-sheet stability.

Observations indicate that the western and southeastern regions have experienced the most significant ice-mass loss in recent decades, under ongoing global warming (Otosaka et al., 2023; Khan et al., 2020; Mouginot et al., 2019). In particular, Jakobshavn Isbrae has been the main contributor to the GrIS-related sea-level rise to date. The outlet glaciers in the northern margin are susceptible to greatly increasing their ice discharge, as their current velocities are slow due to their buttressing
ice shelves (Mouginot et al., 2019). In recent decades, they have experienced a significant increase in their losses (Khan et al., 2022), and if they lose their ice shelves, their contribution to sea-level rise will be similar to that of the western and southeastern regions. Our simulations agree with these observations, illustrating that these areas are the first to show changes with warming (except for the southeast, where losses occur slightly later in our simulations). However, other studies (Höning et al., 2023) suggest a possible equilibrium state in which only the southern sector of the GrIS is lost. They identified two tipping points:
one between +0.6 K and +0.9 K, where only the southern GrIS melts, and another one between +1.6 K and +2 K, where the ice sheet almost completely disappears, with ice remaining only in the northeast. In contrast, in our stability diagram, intermediate states during this transition are only found at +1.4 K and +1.6 K, and they oscillate between 1.51 and 1.97 million km$^3$ and between 0.60 and 1.60 million km$^3$. These oscillations alternate between an intermediate ice-sheet configuration—in which, due to low precipitation in the north and rising temperatures, ice disappears from the northern region—and a near ice-free state.
Therefore, in our case, we do not consider this transition to involve two distinct tipping points.





When relating these values to global mean temperature anomalies with respect to the pre-industrial period in the same way as Bochow et al. (2023), we obtain a tipping point between +1.5 and +2 K, a value within the range reported in some previous studies (Höning et al., 2023; Bochow et al., 2023; Robinson et al., 2012) and below that of others (Noël et al., 2021; Gregory and Huybrechts, 2006). This is a temperature anomaly that will be exceeded during this century unless drastic measures are taken (IPCC, 2021), and even the goal of the Paris agreement to limit global warming to +2 K (+1.5 K if possible) does not guarantee safety McKay et al. (2022).

However, equilibrium states should not be confused with future projections. As we have seen, the transient convergence of the GrIS to its equilibrium states depends on the rate and time of forcing. Nevertheless, these states act as attractors, providing an approximation of the ice sheet's tendency under a given forcing scenario. Thus, if anthropogenic climate change continues and this temperature threshold is exceeded for a sustained period of time, the GrIS could be subject to the largest bifurcation and reach a virtually ice-free state over the next millennia.

## 5   Conclusions

In this study, we perform a stability analysis of the GrIS, incorporating both quasi-equilibrium simulations and equilibrium states. Covering a range of ice-sheet states, from LGM-like to near ice-free conditions, we identify two tipping points: one between -10 and -9 K and the other one between +1.2 K and +1.8 K of regional summer air-temperature anomaly. For the first tipping point, the ocean plays a dominant role in driving ice loss, particularly through basal melting in combination with MISI in some locations. In later stages, as temperatures rise and the GrIS crosses the first tipping point, the diminishing contact between the ice sheet and the ocean shifts the primary drivers of mass loss to atmospheric processes. However, throughout this entire phase between –9 K and +1.2 K, the GrIS shows almost no sensitivity to temperature increases and exhibits a linear behavior with a slope close to zero. Once temperature anomalies exceed +1.2 K, atmospheric feedbacks are triggered and the ice sheet undergoes a drastic reduction, eventually stabilizing at a nearly ice-free state with only residual ice persisting in the highest-altitude regions. Consistent with previous work, this transition suggests that exceeding this temperature threshold under ongoing anthropogenic climate change would result in a virtually ice-free GrIS, with severe implications for global sea-level rise.

The first tipping point has a regional character and only affects the northeast. In contrast, the second one, although beginning at slightly lower temperatures in the west and north in the quasi-equilibrium simulations, has a continental-scale reach. Once this second threshold is exceeded, the GrIS eventually loses practically all of its volume. Conversely, the regrowth of the ice sheet is much slower and requires colder temperatures, illustrating the multistability in the processes of ice loss and recovery. Hysteresis persists across almost the entire temperature range. Its amplitude is greatest between 0 and 1.4 K, due to climate feedbacks.

Throughout this study, we have mapped the stability of the GrIS for temperatures corresponding to the last glacial cycle and those expected over the coming centuries. In this way, two distinct mechanisms affecting its stability—and giving rise to two different bifurcation points—have been identified, further underscoring the importance of considering ice–ocean interactions



in colder climates. The bifurcation point that we found for future anthropogenic warming is within the lower range of previous
415 estimates and even if the Paris Agreement is met, this limit will be surpassed in the coming decades.

*Code availability.* The Yelmo-REMBO coupled model code is available at https://github.com/palma-ice/yelmox and https://github.com/
palma-ice/rembo. The model documentation is available at https://palma-ice.github.io/yelmo-docs/ (last accessed: 30 May 2025).

*Data availability.* The simulations of this work are available on Zenodo (https://doi.org/10.5281/zenodo.15553373 Gutiérrez-González,
2025a).

420 *Video supplement.* The animation of the stability diagram (slowest quasi-equilibrium simulation) can be found on Zenodo (https://zenodo.
org/records/15546141 Gutiérrez-González, 2025b)

## Appendix A: Present-day performance

In the following, we present the comparison of the Yelmo-REMBO simulation for present day conditions to observations (Fig.
A1). As the initial states of the bifurcation diagram (Fig. 1), this third spin-up is initialized with present-day topography and
425 ice thickness (Morlighem et al., 2022), but it is forced with a constant climate corresponding to the average conditions between
1958 and 2001, running for 60 kyr. We obtain a volume of 3.16 million $km^3$, which is 5% higher than present-day observations
(Morlighem et al., 2017). The main discrepancies in the ice-surface elevation (Fig. A1) are in the southwestern margin, where
the ice margin is too advanced, and along the northeastern and eastern coasts, where ice thickness is overestimated. In addition,
this excess ice, which starts from the margins, spreads slightly inland. The overestimation of ice in the north is a common
problem in simulating the GrIS, as noted in several studies (Stone et al., 2010; Robinson et al., 2012; Born and Nisancioglu,
2012; Tabone et al., 2018; Höning et al., 2023). Additionally, REMBO shows an excess in accumulation in the northeast
compared to other regional models (Robinson et al., 2010). And even if a scaling correction to the precipitation has been
applied, small changes in atmospheric forcing can lead to significant changes in ice sheets (Niu et al., 2019). Moreover, the
grid size of 16 km results in misrepresentations of ice-ocean interactions in fjords, where the ocean penetrates the continent
through narrower channels than the grid resolution allows. Therefore, the present day simulation shows ice extending slightly
beyond its present day margin. The same applies in the east, where the topography shows abrupt variations. Regarding surface
ice velocities, our model performs well against the observations. The only exceptions are the Northeast Greenland Ice Stream
(NEGIS), which is generally challenging for ice sheet models to represent, and the innermost regions of glaciers, where our
model reduces velocities too early when moving inland. Even so, this is a very good overall representation for simulations in
which the basal friction has not been optimized.



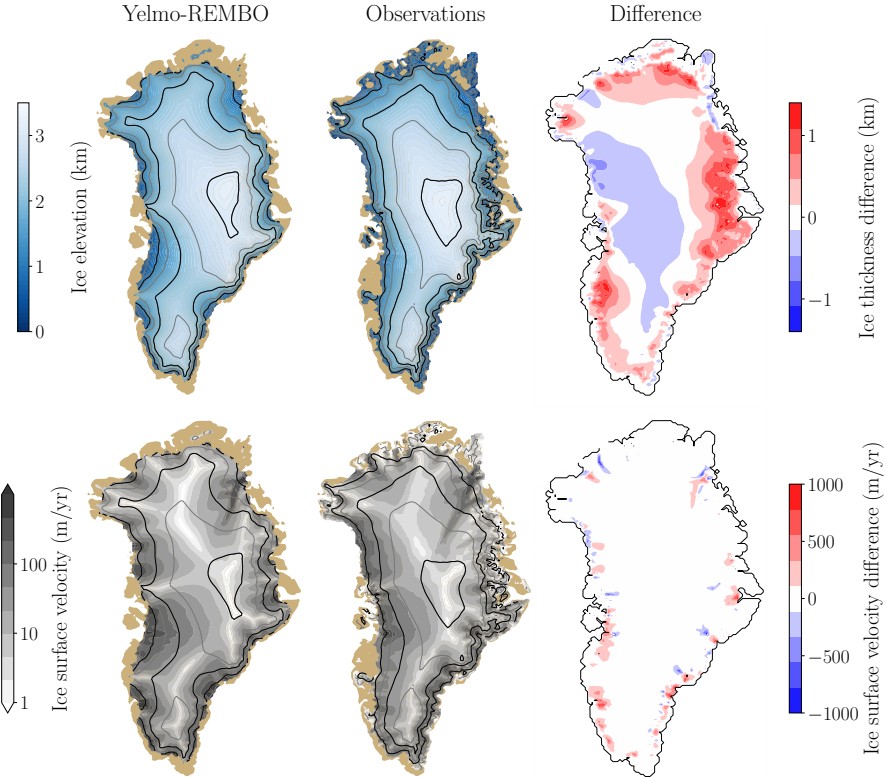

**Figure A1.** Present-day performance. First row: ice-surface elevation, observations from Morlighem et al. (2022) at 16-km resolution. Second row: the ice-surface velocities, observations from from Joughin et al. (2018). The black contour lines indicate the surface elevation every 500 m starting from 0 and with a thicker line every 1000 m.

The aim of this study is not to achieve a perfect representation of the present day ice sheet, but to investigate its stability and thresholds over a broad range of temperatures. The experimental set-up allows both a representation of the present-day and the LGM-like state of the ice sheet which is adequate for this task. The choice of a 16-km grid size means that some processes may not be represented. However, it allows for quasi-equilibrium simulations extending up to a million of model years, which 445 would not be computationally possible with a higher resolution grid.





## Appendix B: Branching-off equilibrium states

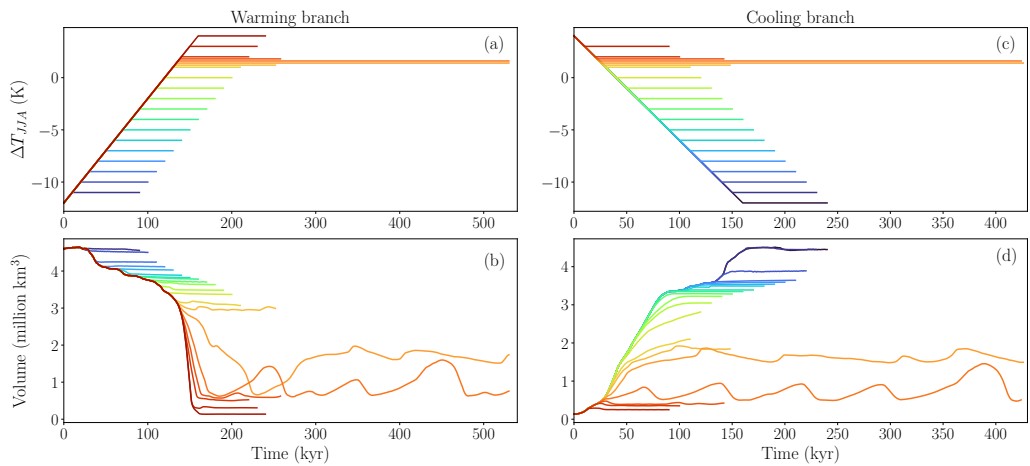

**Figure B1.** Results of the branching-off simulations. Panels a) and b) represent the warming branch, while c) and d) show the cooling branch. The first row displays the forcing: the summer regional air-temperature anomaly, and the second row shows the volume above flotation. Starting from the quasi-equilibrium simulation with a forcing rate of $1 \cdot 10^{-4} \mathrm{K} \cdot \mathrm{yr}^{-1}$, each simulation is extended for an additional 80 kyr after reaching the corresponding temperature level—except for the +1.2 K and +1.8 K simulations, which are extended for 120 kyr, and the 1.4 K and 1.6 K simulations, which are extended for 400 kyr.

## Appendix C: Sensitivity to the basal melting parameters

The first bifurcation point, being triggered by ocean warming, is strongly influenced by the parameters of the basal melting at the grounding line, $\kappa$ and $B_{ref}$, following the equation:

$$B_{gl} = \kappa \Delta T_{ocn} + B_{ref} \tag{C1}$$

As explained in the main text, $\Delta T_{ocn}$ is the ocean temperature anomaly relative to the present-day value, and it is related to the regional summer air temperature anomaly by $\Delta T_{ocn} = 0.25 \Delta T_{ann} = 1.5 \Delta T_{JJA}$. In Fig. C1, it can be seen that as $\kappa$ increases, higher temperatures are required to activate basal melting, allowing both the ice shelves and the northeastern margin to persist at higher temperatures. The opposite occurs with $B_{ref}$: as it increases, the tipping-point value decreases. In these two ensembles, parameter values were chosen to allow the GrIS to grow at LGM conditions while remaining consistent with observations (Wilson et al., 2017). This results in a range for the first bifurcation point between -11 and -8 K, making it highly sensitive to the parameterization of ocean interactions. However, despite the different temperatures at which this bifurcation point occurs, the process is the same across all simulations: the loss of ice shelves followed by an abrupt retreat in the northeast, accelerated by the MISI. In agreement with our mechanistic attribution of the bifurcation points, the second one is not affected by changes in the ocean parameterization.



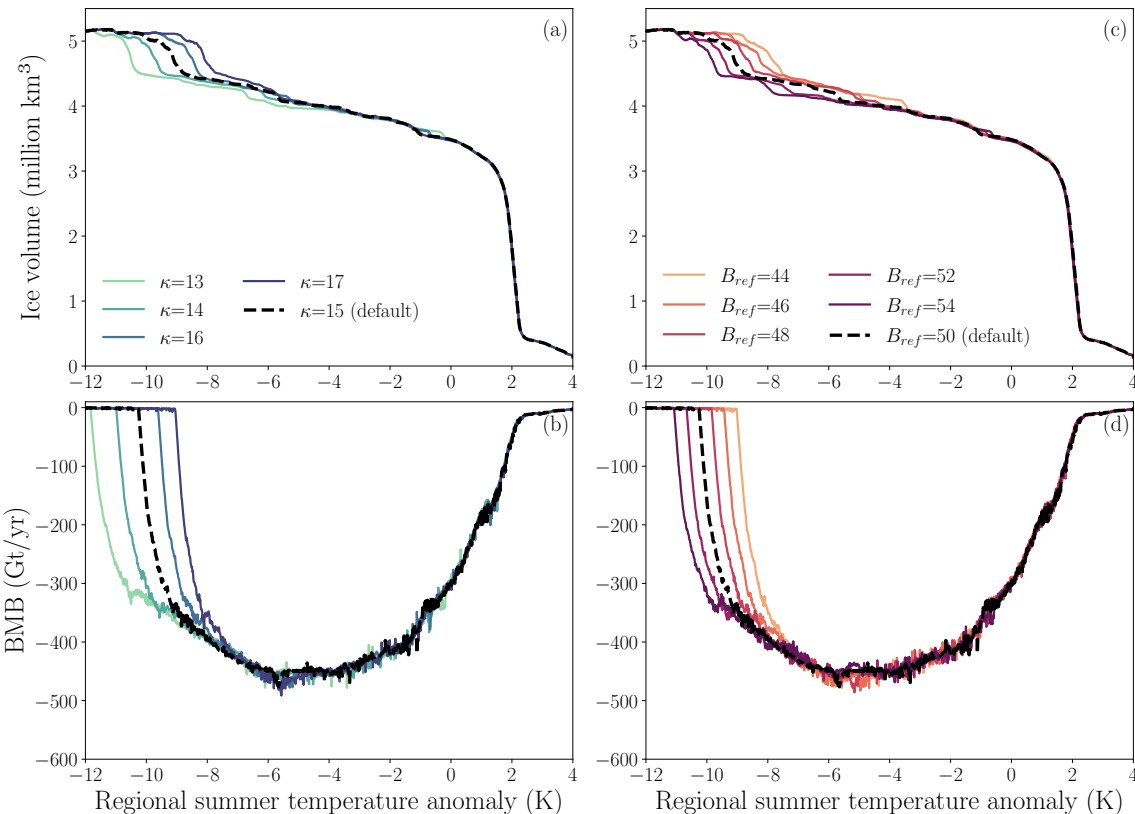

**Figure C1.** Warming branch quasi-equilibrium simulations (rate of forcing of $3 \cdot 10^{-4} \mathrm{K} \cdot \mathrm{yr}^{-1}$) for different values of $\kappa$ and $B_{ref}$. a) The total ice volume (grounded and floating) and b) the basal mass balance.

*Author contributions.* LGG ran all the simulations, analysed the results and wrote the paper. All other authors contributed to the analysis of the results and writing of the paper.

*Competing interests.* AR is member of the editorial board of The Cryosphere.

*Acknowledgements.* LGG is founded by the Comunidad de Madrid. All simulations were performed in Levante, the HPC server of the Deutsches Klimarechentrum GmbH. This is ClimTip contribution #68; the ClimTip project has received funding from the European Union's Horizon Europe research and innovation programme under grant agreement No. 101137601.



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
