# Peer review of "Hysteresis of the Greenland ice sheet from the Last Glacial Maximum to the future"

_EGUsphere, 2025_

## Referee Comment (RC1)

Gutiérrez-Gonzáles et al. study the hysteresis behavior of the Greenland Ice Sheet under a wide range of temperature anomalies, also exploring the effect of cooling for the stability of the Greenland Ice Sheet. They use a state-of the art ice sheet model (Yelmo) and the coupled regional mass balance model REMBO. For the warming of the Greenland Ice Sheet starting at an equilibrium state under temperatures close to the last glacial maximum, the authors find two thresholds for the stability of the Greenland Ice Sheet: starting at an equilibrium with a regional summer air temperature anomaly of -14 K (close to the temperatures of the last glacial maximum) the first threshold is found at temperature anomalies between -10 to -9 K. Here, the marine parts of the ice sheet are lost. The second threshold is found temperature anomalies between +1.2 and +1.8 K and leads to an almost complete loss of the ice. A hysteresis behavior is observed over almost the whole range of temperature anomalies.

The authors analyze both the warming and the cooling branch. They distinguish between oceanic and atmospheric influences, and they undertake a regional study for the warming branch.

**General comments:**

While this is not the first study on the topic of Greenland Ice Sheet stability, it is to my knowledge the first study which takes the full glacial-interglacial temperature range into account. Even so, while the stability of the Greenland Ice Sheet has been studied with different models and different forcings, many details seem to depend on the type of model, the coupling to the atmospheric conditions and the exact type of forcing. Therefore, it is worthwhile adding another piece of information, in my opinion.

The manuscript is generally well written, clear and easy to understand, however, I have a few methodological and logical concerns.

1) The authors claim that the initial state of the warming branch at a temperature anomaly of -14 K is an LGM-like state and compare it to reconstructions of LGM ice extent. However, some important features do not match LGM, particularly the insolation and the sea level. There is a gap between the attempt of an idealized analysis of the stability landscape and the connection to a realistic paleoclimate, which is not easily bridged. I suggest that the authors use more careful language at this point, so that the idea that the "cold" initial state should be equivalent to the actual LGM configuration of the Greenland Ice Sheet doesn't arise. In addition, a sensitivity analysis with LGM levels of insolation or sea level might be beneficial towards comparing the modeled "cold" state with LGM ice extent reconstructions.

2) The authors undertake some efforts to distinguish different feedbacks which influence or even trigger the tipping of the Greenland Ice Sheet. The claim that the first tipping is driven by ice ocean interaction and MISI in the north-west is

well supported by the data in the following subsection. However, the claim in line 267 "Finally, at ~+1.5 K elevation and albedo feedbacks are triggered and the SMB drops abruptly and becomes negative." is not supported by a similarly thorough analysis. The data shown in Figure 6 is not sufficient to support the claim of very specific feedbacks at play. Similarly, in line 270 an increase in sub-shelf melting, a reduction of ice shelves and a margin retreat are mentioned, which isn't visible in Figure 6 either (nor in any other of the presented figures). I appreciate the attempts to disentangle the different mechanisms at work for the two different tipping points. I would suggest strengthening analysis in section 3.2 and backing it with a more thorough analysis of the data.

**Detailed Comments**

L25 and l36: consider citing Solgaard et al. (2012) among the other studies on the topic of Greenland tipping

L79: Calving seems to play some role in the crossing of the tipping points. Therefore, I would appreciate if the Von Mises stress criterion for calving would be motivated a little better and ideally discussed.

L85: please define "present day" with an exact time period

L112: Do I understand correctly that the humidity over the ocean is held constant at the boundary conditions and does not increase with increasing temperatures? Does the precipitation within the simulation box adapt to changing temperatures? How much additional effect would be expected from a humidity correction at the boundaries?

L138: express the equation also in terms of $\Delta T_{JJA}$. I was confused by the numbers if line 155 and line 159 because it slipped my attention that $\Delta T_{ocn}$ refers to the annual temperature anomaly instead of the "normal" input.

L139: "... for purely floating ice shelves..." The sentence is a bit unclear, as all ice shelves are attached to the ice sheet, by definition. Do you mean in simulation cells with purely floating ice shelves? Does the ice model Yelmo have a mechanism for partially floating and partially grounded ice within a simulation cell?

L162: As mentioned above, the comparison to the LGM reconstructions might imply to the reader, that the starting point for the warming branch is indeed an LGM climate. Please clarify this paragraph.

L174: I suppose the mass balance is negative at the margins of the former ice sheet, not the equilibrium state reached at $\Delta T = +4$ K.

Figure 1: Why is the shaded ice not taken into account for the volume calculation?

L239: "These results clearly show the impact of atmospheric feedbacks related to elevation and albedo…" which is not so clear to me from the data. Please clarify or add additional information.

L266ff: This paragraph explains the changes in the surface mass balance, however, all claims in the paragraph are not supported by data which would be available to the reader. This contrasts with the thorough discussion on grounding line shape and ice dynamics on the previous section. Please, support the claims with data or with references.

L298f: I find this sentence surprisingly hard to read. I also kept wondering, why the volume decreases if the tipping point isn't reached yet. Consider rewriting for clarity.

L301f: I'm not sure if the relationship between initial melting and the acceleration of ice flow is sufficiently explained. And I didn't quite understand how the spike in calving is related to the previous.

L328ff: As far as I understand the sea level remains constant in these simulations. How would a decrease of sea level to a realistic number during LGM affect the MISI-driven bifurcation?

L358: Is there any interpretation for the existence of intermediate states? How does this compare to the intermediate states found in Robinson et al. 2012?

L360ff: I understand that further analysis might be beyond the scope of the study. I would still be curious to hear more about why the initial ice volume is only regained at temperatures of at $\Delta T = $ -5 K.

Appendix A: How does the present-day state compare to the equilibrium states at $\Delta T = $ 0 K on the cooling and the warming branch? What does it mean for the stability of the present-day state, that it is in the unstable zone between two stable branches?

**References**

Solgaard, A. M., & Langen, P. L. (2012). Multistability of the Greenland ice sheet and the effects of an adaptive mass balance formulation. *Climate dynamics*, *39*(7), 1599-1612.

Robinson, A., Calov, R., & Ganopolski, A. (2012). Multistability and critical thresholds of the Greenland ice sheet. *Nature Climate Change*, *2*(6), 429-432.

---

## Referee Comment (RC2)

Review of "Hysteresis of the Greenland ice sheet from the Last Glacial Maximum to the future"
by Gutiérrez-González et al.

The manuscript presents a set of simulations of the Greenland ice sheet, spanning from glacial conditions to warm climate states, with hysteresis experiments. The simulation is conducted using the ice sheet model Yelmo, coupled with the energy balance model REMBO. The hysteresis of the Greenland ice sheet is investigated by temperature anomaly through surface mass balance and oceanic basal mass balances. The hysteresis of the Greenland ice sheet is characterised by the wide range of temperature anomalies, spanning from LGM-like glacial states to warm states just before ice-free Greenland. The analysis focusing on specific region of Greenland ice sheets shows that the threshold of the marine-based ice sheet makes the hysteresis of the Greenland ice sheet.

I think this study's topic is well-suited for Climate of the Past. The model is well developed and the experimental design is carefully chosen, and the figures provide good analysis. The manuscript is well-written and clear. However, I identified some areas for revision in the manuscript; therefore, I appreciate the authors addressing several points before publication.

**General comments**
[1] As in the abstract and conclusion, one primary finding is that the hysteresis of the Greenland ice sheet is identified near glacial conditions corresponding to -10K to -9K in regional summer temperature. However, the text does not appear to have adequately considered the assumptions that underlie the derivation of this temperature values. The equation (6) $\Delta T_{ocn} = 0.25\Delta T_{ann}$ is one example. As stated in the main text, this equation uses the same equation as Golledge (2015). However, as the study of Golledge et al. (2015) is for Antarctic ice sheet, it needs some justification arguments why they identified this value. For example, In a more recent study, Garbe et al. (2020) used $\Delta T_{ocn} = 0.39\Delta T_{ann}$ for the Antarctic ice sheet, based on an analysis of the Antarctic region in a 4xCO2 climate model simulation.

I agree with using formulation (6) for all experiments in this paper, but as analyzed in additional experiments (Appendix Figure C1) and L344-348, I believe it is important to note that showing temperature values of -10 to -9 K in the conclusions and abstract contains significant uncertainty in relating atmospheric temperature to ocean temperature.

[2] I couldn't fully understand how ocean temperature works in the experiments. According to Section 2.3, if I put the parameters $\kappa$=15 and Bref=50 into Equation 5, $\Delta$Tocn=-3.33K induces Bgl=0. And using equation (6), Bgl=0 when $\Delta$Tjja=-8.89K, and below that temperature anomaly, Bgl=0 as Bgl cannot be negative (L134).

However, according to Figures 6 and 7, basal mass balance is still significantly greater than 0 even if $\Delta$Tjja is around -10K.

Accorinding to my calculation above, in Figure 8, both $\Delta$Tjja=-9.3K and -9.4K would induce Bgl=0, I'm confused. Maybe I'm doing something wrong.

I have a suggestion. Since Bgl should have a uniform value, I think it would be possible to plot the Bgl values on top of Figure 3. Wouldn't it make it clearer how the basal melting works?

[3] Tabone et al (2018) is one important previous study of this article of this study because the discussion of the evolution of Greenland ice sheet is discussed (final sentence of the abstract). I find there are many improvements and changes in the model setup compared to Tabone et al. (2018). However, the manuscript does not clarify that basal freezing was possible in Tabone et al. (2018), contrary to this study. I like the setup of this study preventing basal freezing, because Antarctic Ocean modeling indicates still active basal melting in the glacial conditions because thermal forcing is positive (Kusahara et al. 2015; Obase et al. 2017). Would it be possible that the presence or absence of basal freezing can have a substantial impact on the hysteresis?

[4] In the experiment, the atmospheric surface mass balance and the ocean basal mass balance change simultaneously in response to $\Delta$Tjja. However, an additional experiment in which one of the forcings is turned off, would identify the mechanism. For example, in the -9.3K experiment (Figure 8), if the tipping point does not occur when only Bgl is set to -9.4K, we can strongly argue that the mechanism of MISI is oceanic forcing.

**Detailed comments**

L4: "global warming to 4K" would be changed to "threshold of ice-free state"?

L7-8: Please clarify that -12 K and +4K indicate regional summer temperature.

L34-35: On the threshold of Greenland ice sheet, recent study (Gregory et al., 2020) addresses this topic, with the effect of ice sheet-climate interactions and the irreversitbility of the Greenland ice sheet.

L41: "regional summer atmospheric temperature" Where? Is it based on ice core site NGRIP?

L80: Is the extent of the ice shelf margin determined only by stress alone? Are there any geographical constraints like continental shelf break positions?

L83-84: As far as I understand, the REMBO needs specific humidity as the input. According to the model results' description, an increase in regional atmospheric air temperature leads to an increase in precipitation over Greenland (e.g., L266). How is the specific humidity treated in temperature changes? Is it assuming the relative humidty as the constant?

L89: "100 km" which model's resolution? I suppose the resolution of ERA-40 is ~100 km; could you please clarify this?

L90: According to Robinson et al. (2010), the REMBO utilized empirical lapse rate feedback of 6.5 K/km for elevation correction. Is the same elevation-temperature feedback utilized in the current model used in these experiments?

L107: Is δP defined at every 16 km grid cell? It would be helpful to put the map of P and Pcorr in the supplemental Figure

L109: What does "consistent field" mean?

L128: Is The unit m/yr defined as "freshwater equivalent" mass balance? Or ice equivalent? Please clarify.

L156: I think retaining insolation as present-day is one probable experimental design because summer insolation in the northern high latitude at the LGM is similar to present-day. However, I recommend the author consider adding one sensitivity experiment setting reduced sea level and setting LGM insolation values (used in the energy balance model REMBO as in equation 3) to assess the impact of these parameters.

L299: Please clarify at what degree of $\Delta T_{jja}$ does basal melting activate?

L349-L354: In Bochow's (2023) experiment, oscillations were not observed in YELMO-

REMBO. However, oscillations were observed in the experiment described in this article. I have identified that the experimental design is not identical with Bochow et al. (2023), the one is the scaling ratio of $\Delta$Tjja and $\Delta$Tdjf (1.61 in Bochow et al. (2023) while the scaling ratio is 2 in this study. Are there other differences in how ocean melt is determined?

I believe it would be beyond scope of this study to explain why oscillatory solutions appear in the experiments in the current setup. I recommend summarizing the differences in the experimental setup compared to Bochow et al. (2023) and stating that the existence of oscillatory solutions depends on the experimental setup.

L373-L380: Please clarify that Honing (2023) defines temperature as the global mean temperature, which differs from this study and Robinson (2012).

L381-L386: Bochow et al. (2023) derives the relationship between global mean temperature and $\Delta$Tjja based on an analysis of the CMIP6 climate model historical and the SSP585 experiment. I recommend summarizing the method of relating global mean temperature and $\Delta$Tjja in the manuscript text.

Figure C1: It would be good to have a diagram with Mgl on the horizontal axis, which would allow us to consider whether the basal mass balance of the ocean or the surface mass balance of the atmosphere primarily determines hysteresis.

**Other minors:**
L5: Yelmo coupled with regional energy balance model REMBO
L47: "regional climate model REMBO" to "regional energy balance model REMBO" to make consistency.
L82: "regional climate model" to "regional energy balance model"
Figure A1: Could you please show the distribution of SMB in the current climate in this experimental setting, with a comparison to SMBMIP (Frettweis et al. 2020)?

**References:**

Kusahara, K., T. Sato, A. Oka, T. Obase, R. Greve, A. Abe-Ouchi, and H. Hasumi (2015), Modelling the Antarctic Marine Cryosphere at the Last Glacial Maximum, Annals of Glaciology, 56(69), 425-435. doi:10.3189/2015AoG69A792.
Obase, T., A. Abe-Ouchi, K. Kusahara, H. Hasumi, R. Ohgaito (2017), Responses of basal melting of Antarctic ice shelves to the climatic forcing of the Last Glacial Maximum and CO2

doubling, Journal of Climate, 30(10), 3473-3497. doi:10.1175/JCLI-D-15-0908.1.

Gregory, J. M., George, S. E., and Smith, R. S.: Large and irreversible future decline of the Greenland ice sheet, The Cryosphere, 14, 4299–4322, https://doi.org/10.5194/tc-14-4299-2020, 2020.

Fettweis, X., and others.: GrSMBMIP: intercomparison of the modelled 1980–2012 surface mass balance over the Greenland Ice Sheet, The Cryosphere, 14, 3935–3958, https://doi.org/10.5194/tc-14-3935-2020, 2020.

---

## Author Comment (AC1)

**Response to Maria Zeitz (RC1)**

We want to thank the reviewer for their effort and the very helpful comments on our manuscript. Below find detailed answers to all comments. The reviewer's comments are in blue, our answers in black and in *italic* we show specific modifications to the original manuscript.

**General comments:**

While this is not the first study on the topic of Greenland Ice Sheet stability, it is to my knowledge the first study which takes the full glacial-interglacial temperature range into account. Even so, while the stability of the Greenland Ice Sheet has been studied with different models and different forcings, many details seem to depend on the type of model, the coupling to the atmospheric conditions and the exact type of forcing. Therefore, it is worthwhile adding another piece of information, in my opinion.

The manuscript is generally well written, clear and easy to understand, however, I have a few methodological and logical concerns.

1) The authors claim that the initial state of the warming branch at a temperature anomaly of -14 K is an LGM-like state and compare it to reconstructions of LGM ice extent. However, some important features do not match LGM, particularly the insolation and the sea level. There is a gap between the attempt of an idealized analysis of the stability landscape and the connection to a realistic paleoclimate, which is not easily bridged. I suggest that the authors use more careful language at this point, so that the idea that the "cold" initial state should be equivalent to the actual LGM configuration of the Greenland Ice Sheet doesn't arise. In addition, a sensitivity analysis with LGM levels of insolation or sea level might be beneficial towards comparing the modeled "cold" state with LGM ice extent reconstructions.

We appreciate your comments regarding the characterization of our "LGM-like" initial state and its comparison with a realistic LGM state. We agree that further clarification would strengthen the paper. For the sake of clarification, the temperature anomaly relative to present-day for our LGM-like state is -12 K of regional summer air temperature, not -14 K.

Your suggestion of a sensitivity analysis with LGM insolation and sea level levels is a valuable addition. We have therefore conducted five new 60-kyr spin-up simulations with a regional summer air temperature anomaly of −12 K, varying insolation and sea level, and revised the manuscript as described below. According to the relative sea level (RSL) reconstruction by Waelbroeck et al. (2002), RSL during the LGM was 120 m below present-day values (±13 m uncertainty). During the subsequent 5 kyr, regional summer air temperatures increased by only ca. 2 K (Buizert et al., 2018), while sea level had risen to 80 m below present-day values. Given this substantial change in boundary conditions with minimal temperature variation, we performed the following experiment permutations under identical temperature forcing as the LGM-like state:

- Present-day (PD) insolation with PD sea level (the simulation presented as the LGM-like state in the original manuscript)
- PD insolation with sea level at -120 m
- PD insolation with sea level at -80 m
- LGM insolation with PD sea level

- LGM insolation with sea level at -120 m
- LGM insolation with sea level at -80 m

**Figure R1.1:** Final state of spin-up simulations under −12 K regional summer air temperature anomaly forcing. The upper row shows simulations with present-day insolation; the lower row shows simulations with 20 kyr BP insolation. The first column represents present-day global sea level conditions, while the second and third columns show simulations with global sea levels 80 m and 120 m below present-day values, respectively. Volume above flotation (in millions km³) is displayed at the bottom of each panel. The black contour lines indicate the surface elevation every 500 m starting from 0 and with a thicker line every 1000 m. The blue lines indicate the ice-sheet margin of Lecavalier et al. (2014) and the green line is the maximum extent of grounded ice from the full glacial extent (18-16 kyr BP) in the northeast region by Leger et al. (2024).

Fig. R1.1 displays the different equilibrium cold states corresponding to each of these six cases under a  $\neg 12$  K regional summer air temperature anomaly forcing. The primary difference lies in the amount of floating ice along the western and southeastern margins. The simulation with reduced insolation and lowest sea level (Fig. R1.1e) most closely approximates LGM boundary conditions. Under these conditions, the majority of previously floating ice is grounded, resulting in the highest volume above flotation: 5.06 million km³, equivalent to an anomaly of 4.81 m of sea level equivalent (SLE)—a value consistent with previous studies.

More importantly, the ice-sheet geometry remains similar across all reconstructions and falls within reconstructed paleoclimatic margins, consistent with the original simulation. Notably, the

northeastern margin, where the first tipping point occurs, exhibits virtually identical behavior across all simulations.

This allows us to address the reviewer's comment presented later:

"L328ff: As far as I understand the sea level remains constant in these simulations. How would a decrease of sea level to a realistic number during LGM affect the MISI-driven bifurcation?"

Indeed, sea level remains constant throughout the equilibrium diagram to isolate the temperature effect on the ice sheet, as explained at line 156. As noted throughout the manuscript, the MISI-driven bifurcation point value is strongly influenced by how ocean-ice interactions are modeled and calibrated, resulting in considerable uncertainty in its precise value. Similarly, we acknowledge that redoing the bifurcation diagram with different sea-level values could potentially alter the bifurcation point. However, given the consistent northeastern margin configuration observed across our LGM-like state simulations, such changes would likely be minor (and inside the ocean forcing uncertainty range illustrated in Fig. C1) rather than substantial modifications to the overall system behavior.

To verify this assessment, we performed a new warming branch of the stability diagram (focusing only on the temperature range around the first bifurcation point) using a quasi-equilibrium simulation (forcing rate of 3·10-5 K·yr-1). This simulation takes the LGM reconstruction from Fig. R1.1f as initial state (LGM insolation and -120 m of sea level). The result is shown in Fig. R1.2, where the main difference compared to the original simulation is a shift in the bifurcation point value (as expected and within the uncertainty range) and a slightly more gradual behavior at the beginning of this transition (around -8.5 K). This second aspect is due to the fact that starting from a slightly larger volume, there is initially a thinning of the ice thickness before MISI is subsequently triggered. The volume difference after crossing the bifurcation point is due to the grounding line remaining slightly more advanced in this simulation than in the original one. These changes in the initial ice-sheet state also generate changes in the bathymetry, which in turn modify the exact equilibrium positions of the grounding line. Nevertheless, we observe that the overall system behavior is preserved. In a transient trajectory from LGM conditions to the present, there would be an increase in temperatures, but also in insolation and global sea level (with the respective contribution from each ice sheet). Therefore, it is reasonable to expect that the exact bifurcation point would lie somewhere between these two idealized states. However, determining the precise transition would be more appropriate for a transient study focused on reconstructing the deglaciation. All in all, figure R.1.2 shows that a decrease of sea level to a realistic number during the LGM does not alter the MISI-driven character of the bifurcation.

**Figure R1.2:** Total ice volume in the North region in quasi-equilibrium simulations with a forcing rate of  $3\cdot10^{-5}$  K·yr-1. The dashed black line shows the original simulation from the manuscript. The blue line shows a new run with LGM insolation and sea level, using the initial state shown in Fig. R1.1f.

Here is the detailed explanation of the changes in the manuscript related with this comment:

- 1. We have included LGM sensitivity analysis (Fig. R1.1) as an appendix in the manuscript.
- 2. We will now talk about "LGM-like state" instead of "LGM state" throughout the text.
- 3. In Section 2.4 (L156), when first introducing the LGM-like state, we will add a more explicit statement: "It is important to note that this LGM-like state is an idealized configuration designed to analyze the effect of temperature on ice-sheet stability in isolation from other major paleoclimate forcings such as changes in sea level, insolation, or atmospheric CO2 that are relevant to achieve a realistic LGM state."

L162: As mentioned above, the comparison to the LGM reconstructions might imply to the reader, that the starting point for the warming branch is indeed an LGM climate. Please clarify this paragraph.

4. In Section 2.4 (L160), when comparing the cold (LGM-like) state with a reconstruction of the LGM our aim was to put in context the initial state of the diagram. Even if lacking important climatic boundary conditions of the LGM, the model achieves a reasonable representation of the state. We will clarify this point in order to avoid confusion.

2) The authors undertake some efforts to distinguish different feedbacks which influence or even trigger the tipping of the Greenland Ice Sheet. The claim that the first tipping is driven by ice ocean interaction and MISI in the north-west is well supported by the data in the following subsection. However, the claim in line 267 "Finally, at ~+1.5 K elevation and albedo feedbacks are triggered and the SMB drops abruptly and becomes negative." is not supported by a similarly thorough analysis. The data shown in Figure 6 is not sufficient to support the claim of very specific feedbacks at play. Similarly, in line 270 an increase in sub-shelf melting, a reduction of ice shelves and a margin retreat are mentioned, which isn't visible in Figure 6 either (nor in any other of the presented figures).

I appreciate the attempts to disentangle the different mechanisms at work for the two different tipping points. I would suggest strengthening analysis in section 3.2 and backing it with a more thorough analysis of the data.

We appreciate the reviewer's observation. We focus our detailed analysis on the first tipping point as it represents a novel finding. In contrast, this second tipping point driven by melt and albedo feedbacks has already been characterized in previous literature (Robinson et al., 2012; Höning et al., 2023; Bochow et al., 2023; Petrini et al., 2025; Pattyn et al., 2018; Noël et al., 2021; Boers et al., 2025), and our results align with these established findings.

We agree that Figure 6 does not explicitly show the different feedbacks; it serves to distinguish the forcing mechanisms that trigger them (differentiating between oceanic and atmospheric drivers). Moreover, these statements cannot rely solely on what figures show. What we can affirm is that the abrupt volume loss preceded by a SMB decline (with the decrease in basal melting, which shows that basal processes play no role in the volume reduction) confirms atmospheric processes as the primary driver. Additionally, at ca. +1.5 K, the abrupt transition from positive to negative SMB values can only be possible with atmospheric feedbacks playing a role. According to previous literature, the feedbacks that trigger this tipping point are albedo and elevation feedbacks, both of which are included in our experiments. Based on this, we can conclude that at ca. +1.5 K these atmospheric feedbacks are activated.

It is true that this description is largely based on the findings of previous studies on this topic, therefore we will modify the text to make this clear. Complementarily, and following also a suggestion from referee #2, we have now explored the stability diagram also by considering the oceanic and atmospheric forcings separately. Figure R2.2 (in the referee #2 response) shows the former hysteresis diagram together with the OCN only and ATM only new realizations. It clearly shows that the second tipping point can only be simulated in presence of the atmospheric feedbacks described above.

**Figure R1.3:** Ice area (grounded and floating) in the warming branch of the quasi-equilibrium simulation of  $1 \cdot 10^{-5} \text{K·yr}^{-1}$  in the temperature range of the first bifurcation point.

**Figure R1.4:** a)-e) Surface mass balance and f)-j) basal mass balance at different temperatures from the warming branch of the quasi-equilibrium simulation of  $1 \cdot 10^{-5} \text{K} \cdot \text{yr}^{-1}$  before the first tipping point. The black contour lines indicate the surface elevation every 500 m starting from 0 and with

a thicker line every 1000 m. Note that the higher values in basal melting in f)-j) highlight the grounding line.

Regarding line 270, we will add Figs. R1.3 and R1.4 to an appendix in the manuscript. Fig. R1.3 shows the grounded and floating area in the warming branch of the quasi-equilibrium simulation of  $1\cdot10^{-5}$ K·yr-1 (black line in Fig. 3a in the manuscript). There, we can see that after a very small loss in grounded ice, there is a significant loss of floating ice, followed by an abrupt retreat of grounded ice (and the grounding line). In Fig. R1.4 we see the evolution of the ice-sheet surface and basal mass balance before the first tipping point (Fig. R1.4a-e and f-i) and just after (Fig. R1.4e and j). We can see that there is basal melting at the grounding line, at the base of the ice shelves and in some regions under the ice sheet where there is a water layer. As temperatures increase, basal melting also increases both at the grounding line around the ice-sheet margin and below the ice shelves. This causes the retreat of the ice shelves until they remain only in reduced form in the southeast. Finally, the melting becomes so high that the northeast retreats abruptly.

**Detailed Comments**

L25 and l36: consider citing Solgaard et al. (2012) among the other studies on the topic of Greenland tipping

This is a good suggestion, thank you. We will add Solgaard and Langen (2012) in this two sentences in the introduction:

"Ice-sheet modelling studies furthermore suggest that the GrIS shows multistability and hysteresis with respect to the temperature forcing (Solgaard and Langen, 2012; Robinson et al., 2012; Höning et al., 2023)."

"In addition, the hysteresis behavior of the GrIS has only been studied under temperatures above present-day values (Solgaard and Langen, 2012; Robinson et al., 2012; Bochow et al., 2023; Höning et al., 2023)."

L79: Calving seems to play some role in the crossing of the tipping points. Therefore, I would appreciate if the Von Mises stress criterion for calving would be motivated a little better and ideally discussed.

We chose this approach because it is physically based on the principle that calving is governed by the tensile stress regime at the ice-sheet front and has demonstrated good performance in reproducing observed calving rates across different glaciers (Morlighem et al., 2016; Choi et al., 2018; Goelzer et al., 2017). While other calving criteria exist that represent the calving phenomenon with greater complexity, the computational cost of applying these methods at the full ice-sheet scale makes it preferable to employ simpler criteria such as Von Mises, which nonetheless yields satisfactory results consistent with previous studies of the GrIS using Yelmo (Tabone et al., 2024).

We will add this motivation to the revised manuscript.

**L85: please define "present day" with an exact time period**

We define the present-day conditions as the average conditions between 1958 and 2001. However, it seems that in one version of the drafts this was mistakenly omitted from the sentence, so we thank the reviewer for noticing it and we have now added it on line 85:

"The temperature and humidity over the ocean around Greenland are imposed as boundary conditions, for which the climatological mean from the years 1958–2001 of the ERA-40 reanalysis (Uppala et al., 2005) to represent present-day conditions."

L112: Do I understand correctly that the humidity over the ocean is held constant at the boundary conditions and does not increase with increasing temperatures? Does the precipitation within the simulation box adapt to changing temperatures? How much additional effect would be expected from a humidity correction at the boundaries?

No, the humidity at the boundary conditions is not constant. Over the ocean, two variables are prescribed: air temperature T and relative humidity r. Then the specific humidity Q at the boundary is given by:

$$Q = Q_{sat}(T) \cdot r$$
;

where  $Q_{sot}(T)$  is the saturation specific humidity, which depends on temperature following the Clausius-Clapeyron equation. Therefore, across the diagram, the temperature at the boundaries changes and the relative humidity is constant, but the specific humidity changes according to the temperature variations. In REMBO, the choice to maintain constant relative humidity rather than specific humidity is based on the fact that specific humidity has a stronger temperature dependence.

The total amount of precipitation is then calculated as a function of the specific humidity, thus accounting for the changing temperature and following the equation:

$$P = (1 + k |\nabla z_{s}|) \left(\frac{Q}{\tau}\right).$$

where  $\nabla z_s$  is the gradient of the surface elevation,  $\tau$  is the water turnover time in the atmosphere and k is an empirical parameter (Robinson et al., 2010). Therefore, even if the relative humidity is constant, the total precipitation changes with temperature. We will clarify this in the manuscript.

We implement a spatially heterogeneous bias correction approach for precipitation to address the systematic biases in the simulated precipitation patterns produced by REMBO, which exhibits enhanced precipitation amounts in the southwestern and northern regions of the domain (Robinson et al., 2010). On the other hand, implementing a boundary humidity adjustment would effectively

modify the total amount of available humidity throughout the domain. However, such an approach would fail to address the underlying spatial heterogeneity in the precipitation bias patterns, whose origin lies in the regional character of the model and its coarse spatial resolution.

L138: express the equation also in terms of  $\Delta TJJA$ . I was confused by the numbers if line 155 and line 159 because it slipped my attention that  $\Delta Tocn$  refers to the annual temperature anomaly instead of the "normal" input.

Fixed.

L139: "... for purely floating ice shelves..." The sentence is a bit unclear, as all ice shelves are attached to the ice sheet, by definition. Do you mean in simulation cells with purely floating ice shelves? Does the ice model Yelmo have a mechanism for partially floating and partially grounded ice within a simulation cell?

When modeling the oceanic forcing, we distinguish between the grid cells located at the grounding line and the rest of the ice shelf. We understand that the original sentence may have been confusing (we were using "purely floating ice shelves" to differentiate the corresponding cells from those at the grounding line), so we have rewritten it as follows:

"On the other hand, the sub-shelf basal melting rate  $B_{sh}$  is lower than that at the grounding line."

L162: As mentioned above, the comparison to the LGM reconstructions might imply to the reader, that the starting point for the warming branch is indeed an LGM climate. Please clarify this paragraph.

Following general comment 1, we have revised the text to prevent confusion as outlined in our previous response.

L174: I suppose the mass balance is negative at the margins of the former ice sheet, not the equilibrium state reached at  $\Delta T$  = +4 K.

The mass balance is indeed negative at the margins of the equilibrium state at  $\Delta T_{JJA} = +4$  K. We see that the original formulation may be ambiguous given the minimal ice extent at this temperature anomaly. However, persistent ice caps remain in the easternmost region and at the southern tip under these conditions. Our reference to negative marginal mass balance refers to these residual ice masses. We have revised the sentence at line 174 as follows:

"The mass balance of the remnant ice caps is almost zero in their interior (all incoming accumulation is melted away) and negative at their margins, especially in areas in contact with the ocean (not shown)."

**Figure 1: Why is the shaded ice not taken into account for the volume calculation?**

We have also simulated the ice evolution on Ellesmere Island due to its influence on the size, shape, and dynamics of the GrIS. However, when calculating the volume of the GrIS, ice on this island is typically not included, both for present-day and past conditions. Therefore, although we simulated ice evolution there as well, we excluded it from our calculations in order to maintain consistency with the domain that is normally included in GrIS volume estimates.

In order to make this clearer in the text, we modify the final sentence in the Figure 1 caption:

"Note that the lightly shaded area in the northwest (over Ellesmere Island) indicates the part of the simulated ice sheet that is not taken into account for the volume calculation in order to be consistent with the usual GrIS domain."

L239: "These results clearly show the impact of atmospheric feedbacks related to elevation and albedo..." which is not so clear to me from the data. Please clarify or add additional information.

See below.

L266ff: This paragraph explains the changes in the surface mass balance, however, all claims in the paragraph are not supported by data which would be available to the reader. This contrasts with the thorough discussion on grounding line shape and ice dynamics on the previous section. Please, support the claims with data or with references.

We will revise these two comments (L239 and L266) in order to clarify that the statements are supported by different studies showing that the albedo and elevation feedbacks generate the presence of this bifurcation point for a warming above present-day values, such as Robinson et al. (2012), Höning et al. (2023), Bochow et al. (2023), Petrini et al. (2025), Pattyn et al. (2018), and Noël et al. (2021). Please see also our answer to general comment # 2.

L298f: I find this sentence surprisingly hard to read. I also kept wondering, why the volume decreases if the tipping point isn't reached yet. Consider rewriting for clarity.

Volume reductions in the western and eastern regions show linear behavior due to the absence of bifurcation points in these areas. Only the northern region exhibits threshold behavior due to its particular bathymetry. Ice losses in the eastern and western regions result from the warming and the increase in BMB and start earlier due to floating ice at their margins, which experiences a higher BMB. The original formulation of the sentence may have been misleading regarding the absence of tipping points in these regions. We rewrite the sentence in this way in the manuscript in order to make it clearer:

"At approximately -10.5 K, the ocean warming triggers the sub-shelf melting, leading to a gradual volume decline. Volume losses are concentrated in the western and southeastern regions, where the ice at the margin is floating and exhibits a higher sensitivity to oceanic forcing at lower temperatures. While these zones experience nearly linear losses up to approximately +2 K, in the northern region—where the ice is grounded and its thickness is higher—the volume remains nearly constant until the temperature anomaly reaches -9.4 K (in this quasi-equilibrium simulation), when the abrupt ice loss occurs."

L301f: I'm not sure if the relationship between initial melting and the acceleration of ice flow is sufficiently explained. And I didn't quite understand how the spike in calving is related to the previous.

Starting from a quasi-equilibrium state, an increase in basal melting generates an imbalance that causes grounding-line retreat. Given the retrograde bedrock configuration, this results in an increase in cross-sectional area, which generates enhanced ice flux towards the exterior. This triggers a MISI, with consequent grounding line retreat, transition of ice from grounded to floating conditions, progressive thinning that finally leads to calving, resulting in a substantial reduction in ice volume. Therefore, the increased calving represents a consequence of the MISI-driven ice flux enhancement. We have reformulated the text to provide a more detailed explanation.

L328ff: As far as I understand the sea level remains constant in these simulations. How would a decrease of sea level to a realistic number during LGM affect the MISI-driven bifurcation?

Addressed above. See our answer to general comment #2.

L358: Is there any interpretation for the existence of intermediate states? How does this compare to the intermediate states found in Robinson et al. 2012?

Conceptually, these intermediate states differ from those reported by Robinson et al. (2012), where the intermediate branch corresponded to the new equilibrium states reached by starting with intermediate (transient) initial conditions. In their framework, the upper branch (referred to in this work as the warming or retreating branch) represents the equilibrium states at different temperatures starting from an initial state similar to pre-industrial conditions. The lower branch (referred to here as the cooling or regrowth branch) represents equilibrium states at different temperatures when starting from a virtually ice-free initial state. The intermediate branch accounts for equilibrium states reached starting from intermediate initial conditions, with the light red area in Figure 1 of Robinson et al. (2012) indicating the volume range of initial states that, when subjected to different temperatures, ultimately reach equilibrium on the intermediate branch.

In contrast, in our simulations, the intermediate states belong to the cooling (lower) branch, meaning that they are reached starting from an ice-free initial state. The key difference from Robinson et al. (2012) is that their results show abrupt regrowth requiring lower temperatures, whereas our

simulations demonstrate regrowth beginning at higher temperatures and proceeding through a two-step process. Nevertheless, it is interesting to highlight that the intermediate state in both studies has a similar shape (covering south and central Greenland) and volume (in relative terms).

This demonstrates that the combination of albedo reduction following considerable ice retreat and low precipitation in northern Greenland makes it more difficult for this region to recover the ice.

Therefore, the existence of these intermediate states in our work is attributed to two factors: first, the stability of an ice-sheet configuration where only southern and central Greenland remain ice-covered (in agreement with Robinson et al., 2012), and second, the capacity of the ice sheet to recover at higher temperatures. To fully understand the reason for this earlier recovery, a comparative analysis between the results of different studies that have examined the hysteresis of the GrIS would be necessary, including factors such as surface mass balance, bedrock and surface elevation, among others. However, we think that such a detailed comparison is beyond the scope of this study.

We will expand the discussion related to these intermediate states in order to make it clearer.

L360ff: I understand that further analysis might be beyond the scope of the study. I would still be curious to hear more about why the initial ice volume is only regained at temperatures of at  $\Delta T$  = -5 K.

If we understand correctly, with "initial ice volume" the reviewer is referring to the ice volume at  $\Delta T_{jja}$  = 0 K (3.4 million km³) in the warming branch. As illustrated in the figure below and Fig. R1.5 of the manuscript, the warming branch equilibrium state at  $\Delta T_{jja}$  = 0 K (hereafter W0) has indeed the same total volume as the cooling branch equilibrium state at  $\Delta T_{jja}$  = -5K (hereafter C-5). However, these states don't have the same ice distribution. The W0 state shows more ice in the NEGIS area and a complete ice coverage over Scoresby Sund, while the C-5 state has a higher ice thickness along the western margin, resulting in equivalent total volumes despite different geometries.

We attribute the delayed volume recovery (requiring cooling to -5K) to the same physical mechanisms responsible for the existence of hysteresis in the -9K to 0K temperature range (lines L245-260). Specifically: (1) coastal margin irregularities prevent ice expansion in certain coastal regions under the C-5 conditions, and (2) bathymetric peak in the NEGIS region acts as a pinning point in the W0 state, allowing ice accumulation. On the other hand, the lower temperatures in C-5 state allow ice thickening in areas with more regular coastal geometry.

Thus, while these states have similar total volumes, they have a different configuration and the volume equivalence is somehow fortuitous. The need for such pronounced cooling to achieve a similar ice-sheet volume underscores the influence of the non-linear feedback mechanisms previously outlined.

**Figure R1.5**: From left to right: warming branch equilibrium state for  $\Delta T_{jja} = 0$  from the branching-off experiment (W0); cooling branch equilibrium state from the branching-off experiment at  $\Delta T_{jja} = -5K$  (C-5); and ice thickness difference between C-5 and W0.  $V_T$  is in million km³ and represents the ice-sheet total volume of each state. The black contour lines indicate the surface elevation every 500 m starting from 0 and with a thicker line every 1000 m.

Appendix A: How does the present-day state compare to the equilibrium states at  $\Delta T = 0$  K on the cooling and the warming branch? What does it mean for the stability of the present-day state, that it is in the unstable zone between two stable branches?

This is a very good question. The present-day state shown in Appendix A has a volume of 3.16 million km³, which falls between the volume of the equilibrium state in the warming branch (3.37 million km³) and the equilibrium state in the cooling branch (2.81 million km³) for that same temperature forcing ( $\Delta T_{ija} = 0$ ). Moreover, when comparing the ice sheet in these three simulations (Fig. R1.6), the present-day state is in between the two equilibrium branches.

Different studies have pointed out that the GrIS is not in equilibrium neither at present nor during pre-industrial times (Yang et al., 2022). This study suggests that the current state results from the ice sheet having retreated beyond its present margin during the Holocene Thermal Maximum (when temperatures exceeded current levels) and subsequently regrowing. Our results also indicate that the present state of the GrIS is not the product of gradual and slow temperature changes since the LGM, as would be the case for the equilibrium states shown in the hysteresis diagram. Instead, achieving the current state starting at the LGM requires an overshooting of present temperatures followed by subsequent cooling, which would have caused the GrIS to transition between the two equilibrium branches since the LGM as it happens during the last deglaciation. We have also considered this question and we plan to address it in future work, including transient simulations of the last deglaciation (starting from a realistic LGM state, actually). Therefore, we do not elaborate on this topic in detail in the present manuscript.

**Figure R1.6**: From left to right: warming branch equilibrium state from the branching-off experiment; present-day steady-state simulation (described in Appendix A in the manuscript); cooling branch equilibrium state from the branching-off experiment. The black contour lines indicate the surface elevation every 500 m starting from 0 and with a thicker line every 1000 m.

**References**

Boers, N., Liu, T., Bathiany, S., Ben-Yami, M., Blaschke, L. L., Bochow, N., ... & Smith, T. (2025). Destabilization of Earth system tipping elements. *Nature Geoscience*, 1-12.

Buizert, C., Keisling, B. A., Box, J. E., He, F., Carlson, A. E., Sinclair, G., & DeConto, R. M. (2018). Greenland-wide seasonal temperatures during the last deglaciation. *Geophysical Research Letters*, 45(4), 1905-1914.

Choi, Y., Morlighem, M., Wood, M., and Bondzio, J. H.: Comparison of four calving laws to model Greenland outlet glaciers, The Cryosphere, 12, 3735–3746, https://doi.org/10.5194/tc-12-3735-2018, 2018.

Goelzer, H., Robinson, A., Seroussi, H., & Van De Wal, R. S. (2017). Recent progress in Greenland ice sheet modelling. *Current climate change reports*, *3*(4), 291-302.

Lecavalier, B. S., Milne, G. A., Simpson, M. J., Wake, L., Huybrechts, P., Tarasov, L., ... & Larsen, N. K. (2014). A model of Greenland ice sheet deglaciation constrained by observations of relative sea level and ice extent. *Quaternary Science Reviews*, *102*, 54-84.

Leger, T. P., Clark, C. D., Huynh, C., Jones, S., Ely, J. C., Bradley, S. L., ... & Hughes, A. L. (2023). A Greenland-wide empirical reconstruction of paleo ice-sheet retreat informed by ice extent markers: PaleoGrIS version 1.0. *Climate of the Past Discussions*, 2023, 1-97.

Robinson, A., Calov, R., & Ganopolski, A. (2010). An efficient regional energy-moisture balance model for simulation of the Greenland Ice Sheet response to climate change. *The Cryosphere*, 4(2), 129-144.

Waelbroeck, C., Labeyrie, L., Michel, E., Duplessy, J. C., Mcmanus, J. F., Lambeck, K., ... & Labracherie, M. (2002). Sea-level and deep water temperature changes derived from benthic foraminifera isotopic records. *Quaternary science reviews*, *21*(1-3), 295-305.

Yang, H., Krebs-Kanzow, U., Kleiner, T., Sidorenko, D., Rodehacke, C. B., Shi, X., ... & Lohmann, G. (2022). Impact of paleoclimate on present and future evolution of the Greenland Ice Sheet. *Plos one*, *17*(1), e0259816.

Bochow, N., Poltronieri, A., Robinson, A., Montoya, M., Rypdal, M., & Boers, N. (2023). Overshooting the critical threshold for the Greenland ice sheet. *Nature*, *622*(7983), 528-536.

Petrini, M., Scherrenberg, M. D., Muntjewerf, L., Vizcaino, M., Sellevold, R., Leguy, G. R., ... & Goelzer, H. (2025). A topographically controlled tipping point for complete Greenland ice sheet melt. *The Cryosphere*, *19*(1), 63-81.

Pattyn, F., Ritz, C., Hanna, E., Asay-Davis, X., DeConto, R., Durand, G., ... & Van den Broeke, M. (2018). The Greenland and Antarctic ice sheets under 1.5 C global warming. *Nature climate change*, 8(12), 1053-1061.

Noël, B., Van Kampenhout, L., Lenaerts, J. T. M., Van de Berg, W. J., & Van Den Broeke, M. R. (2021). A 21st century warming threshold for sustained Greenland ice sheet mass loss. *Geophysical Research Letters*, 48(5), e2020GL090471.

Höning, D., Willeit, M., Calov, R., Klemann, V., Bagge, M., & Ganopolski, A. (2023). Multistability and transient response of the Greenland ice sheet to anthropogenic CO2 emissions. *Geophysical Research Letters*, *50*(6), e2022GL101827.

Robinson, A., Calov, R., & Ganopolski, A. (2012). Multistability and critical thresholds of the Greenland ice sheet. *Nature Climate Change*, *2*(6), 429-432.

Robinson, A., Calov, R., & Ganopolski, A. (2010). An efficient regional energy-moisture balance model for simulation of the Greenland Ice Sheet response to climate change. *The Cryosphere*, 4(2), 129-144.

---

## Author Comment (AC2)

**Response to Takashi Obase (RC2)**

We want to thank the reviewer for their effort and the very helpful comments on our manuscript. Below find detailed answers to all comments. The reviewer's comments are in blue, our answers in black and in *italic* we show specific modifications to the original manuscript.

**General comments**

[1] As in the abstract and conclusion, one primary finding is that the hysteresis of the Greenland ice sheet is identified near glacial conditions corresponding to -10K to -9K in regional summer temperature. However, the text does not appear to have adequately considered the assumptions that underlie the derivation of this temperature values. The equation (6)  $\Delta$ Tocn = 0.25 $\Delta$ Tann is one example. As stated in the main text, this equation uses the same equation as Golledge (2015). However, as the study of Golledge et al. (2015) is for Antarctic ice sheet, it needs some justification arguments why they identified this value. For example, In a more recent study, Garbe et al. (2020) used  $\Delta$ Tocn = 0.39 $\Delta$  Tann for the Antarctic ice sheet, based on an analysis of the Antarctic region in a 4xCO2 climate model simulation.

I agree with using formulation (6) for all experiments in this paper, but as analyzed in additional experiments (Appendix Figure C1) and L344-348, I believe it is important to note that showing temperature values of -10 to -9 K in the conclusions and abstract contains significant uncertainty in relating atmospheric temperature to ocean temperature.

[2] I couldn't fully understand how ocean temperature works in the experiments. According to Section 2.3, if I put the parameters K=15 and Bref=50 into Equation 5,  $\Delta$ Tocn=-3.33K induces Bgl=0. And using equation (6), Bgl=0 when  $\Delta$ Tjja=-8.89K, and below that temperature anomaly, Bgl=0 as Bgl cannot be negative (L134). However, according to Figures 6 and 7, basal mass balance is still significantly greater than 0 even if  $\Delta$ Tjja is around -10K.

According to my calculation above, in Figure 8, both  $\Delta$ Tjja=-9.3K and -9.4K would induce Bgl=0, I'm confused. Maybe I'm doing something wrong. I have a suggestion. Since Bgl should have a uniform value, I think it would be possible to plot the Bgl values on top of Figure 3. Wouldn't it make it clearer how the basal melting works?

We answer these two first comments together since they are closely related. First, we want to acknowledge that the calculations performed in comment [2] are correct. Thanks to this, we have realized that there was an error in the manuscript regarding the scaling factor value between oceanic and atmospheric temperature anomalies that remained from an earlier draft.

The actual value used is 0.22, so that with  $\Delta T_{ann}=1.5\Delta T_{jja}$  we have:  $\Delta T_{ocn}=0.22\Delta T_{ann}=0.33\Delta T_{jja}$ . In this way, for K=15 myr-1K-1 and Bref=50myr-1,  $B_{gl}=0$  when  $\Delta T_{ocn}=-3.33K$  and  $\Delta T_{jja}=-10.1K$ . That is why at around -10K of summer regional temperature anomaly, the basal melting is higher than 0. We thank the reviewer very much for bringing this mistake to our attention.

We will explicitly add this basal melting activation temperature ( $\Delta T_{jja} = -10.1$  K;  $\Delta T_{ocn} = -3.3$  K) to the revised version of the text. However, we do not consider it necessary to include  $B_{gl}$  in Fig. 3 of the manuscript, as Fig. 6 already shows the mass flux information, including total basal melting, and reflects the timing of its activation.

Regarding comment [1], the scaling factor between oceanic and annual atmospheric temperature  $(f = \frac{\Delta T_{ocn}}{\Delta T_{atm,annual}})$  is indeed highly uncertain. Garbe et al. (2020) used f = 0.39 based on the value obtained for a four-fold increase in CO2 in Antarctica once equilibrium was reached with the ECHAM5–MPIOM coupled model. Golledge et al. (2015) used f = 0.25 based on the CMIP5 multi-model ensemble mean in Antarctica. However, both studies are based on inferences from the Antarctic domain and for temperatures above present-day values.

We calculated the mean ratio between oceanic and atmospheric temperature over the ocean surrounding Greenland using PMIP3 models outputs (CCSM4, CNRM-CM5, FGOALS-g2, IPSL-CM5A-LR, MIROC-ESM, MPI-ESM-P, MRI-CGCM3). This yields  $f_{LGM}=0.18\pm0.02$  for the LGM and  $f_{Hol}=0.5\pm0.2$  for the mid-Holocene. These values show considerable spread. We chose f=0.22 as it falls within this range and it is closer to LGM values, which is appropriate since ocean forcing has greater influence in the temperature range we investigate. We will discuss this in the main text.

Importantly, exploring the effect of  $\kappa$  in Eq. (5) for basal melting is equivalent to exploring the uncertainty in f, since this equation can also be written as:  $B_{al} = \kappa f \Delta T_{ann} + B_{Ref}$ . As shown in

Appendix C, this has a direct impact on the bifurcation point value. Higher values of f (like higher values of K) shift the bifurcation point toward higher temperatures (note for the first bifurcation point  $\Delta T_{jja} < 0$ , therefore for higher K Bgl = 0 is reached for lower-amplitude anomalies in absolute value). This value is therefore highly sensitive to the experimental setup (as the reviewer is pointing out), which underscores the importance of the uncertainty analysis presented in Appendix C and is the reason why we agree with the reviewer's suggestion. We will accordingly expand our discussion on this effect by acknowledging the associated uncertainty of the ocean-to-atmosphere fraction, f.

Finally, given the large uncertainties in the basal melting, we agree that there are substantial uncertainties also associated with the value of the first bifurcation point. This is what we aimed to show in Fig. C1. We agree with the reviewer's suggestion to put less emphasis on the exact value of the bifurcation point. We will therefore clarify this in the main text, abstract, and conclusions.

[3] Tabone et al (2018) is one important previous study of this article of this study because the discussion of the evolution of Greenland ice sheet is discussed (final sentence of the abstract). I find there are many improvements and changes in the model setup compared to Tabone et al. (2018). However, the manuscript does not clarify that basal freezing was possible in Tabone et al. (2018), contrary to this study. I like the setup of this study preventing basal freezing, because Antarctic Ocean modeling indicates still active basal melting in the glacial conditions because thermal forcing is

positive (Kusahara et al. 2015; Obase et al. 2017). Would it be possible that the presence or absence of basal freezing can have a substantial impact on the hysteresis?

Yes, the fact that refreezing was allowed in Tabone et al. (2018) is indeed a difference from the present study, and we will make this explicit in the text. Eliminating refreezing (by limiting basal mass balance to zero when temperature anomalies would lead to negative values) was an improvement introduced in subsequent papers (Tabone et al., 2019a; Tabone et al., 2019b; Tabone et al., 2024). While local variations in basal melting and refreezing can exist at the ice-shelf base, given our simplification of applying spatially homogeneous melting, it makes more sense to fully eliminate refreezing in the entire domain rather than allow an unrealistic amount of it. This is particularly true in light of the references cited by the reviewer (Kusahara et al., 2015; Obase et al., 2017).

Regarding whether refreezing would affect the hysteresis, evidence from previous papers (Álvarez-Solas et al., 2017; see the reviewing discussion) suggests that allowing for refreezing could enable faster regrowth in a transient run in response to sufficiently low temperatures. However, concerning the equilibrium stability diagram, refreezing would only be activated in the range of summer regional temperature anomalies of approximately -12 to -10 K, when the cooling branch recovers the LGM-like state, thus not substantially affecting the hysteresis or the main conclusions of our study.

[4] In the experiment, the atmospheric surface mass balance and the ocean basal mass balance change simultaneously in response to  $\Delta$ Tjja. However, an additional experiment in which one of the forcings is turned off, would identify the mechanism. For example, in the -9.3K experiment (Figure 8), if the tipping point does not occur when only Bgl is set to - 9.4K, we can strongly argue that the mechanism of MISI is oceanic forcing.

We want to thank the reviewer for this suggestion. As shown in Figure 6 of the manuscript, at this first bifurcation point only basal melting was increasing significantly (since surface mass balance remains nearly constant), making it straightforward to attribute the triggering of the MISI to oceanic processes. Nevertheless, we agree that the additional experiments suggested by the reviewer would provide final confirmation of the respective roles of atmospheric and oceanic forcing in triggering the different feedbacks along the stability diagram. Therefore, we have simulated two quasi-equilibrium experiments with a forcing rate of  $3\cdot10^{-5}$  K·yr-1 under the following conditions:

- Only OCN:  $\Delta T_{jja}$  remains constant (at -12 K during the warming branch and +4 K during the cooling branch, corresponding to initial state values).
- Only ATM:  $\Delta T_{ocn}$  remains constant (at -4 K during the warming branch and +1.3 K during the cooling branch, corresponding to initial state values).

The initial state of both experiments is the same as in the original stability diagram (Fig. 1 in the manuscript).

We selected the same forcing rate as in on our sensitivity analysis of basal melting parameters, as it allows us to maintain simulations close to equilibrium while reducing computational costs compared to slower forcing rates (these experiments require 533 kyr versus 1.6 Myr for the quasi-equilibrium

simulation with a forcing rate of  $1\cdot10^{-5}$  K yr-1). The results are presented in the Fig. R2.1, which shows both branches of the stability diagram for the OCN-only and ATM-only experiments, as well as the default quasi-equilibrium simulation with both atmospheric and oceanic forcings active.

Regarding the warming branch, the OCN-only simulation follows the control run almost exactly until  $\Delta T_{\text{ocn}}$  = -1K, when the margin retreats to the coastline and reaches a constant state regardless of the increase in ocean temperature. There is no second tipping point in this simulation, which implies the latter was due to the atmospheric feedbacks. Reciprocally, the ATM-only simulation exhibits minimal changes until  $\Delta T_{jja}$  = -3K ( $\Delta T_{ocn}$  = -1K). At around  $\Delta T_{jja}$  = -1K the ice sheet loses the northeast sector, then retreats to the present-day margin and subsequently reaches a virtually ice-free state.

Regarding the cooling branch, as expected, in the OCN-only experiment the ice sheet remains in its virtually ice-free state, given that although oceanic temperatures decrease, it is impossible for the ice sheet to grow without a decrease in atmospheric temperatures. In contrast, the ATM-only experiment follows the control run almost exactly until  $\Delta T_{jja} = -7K$ , when in the control run the ice sheet recovers the marine sectors and in the ATM-only experiment the warm ocean doesn't allow the grounding line to advance.

These results confirm the findings indicated in Figure 6 of the manuscript, that ocean warming is triggering the MISI in the northeast, while atmospheric warming triggers the elevation and albedo feedbacks. In the absence of ocean forcing, there is also an abrupt loss in the northeast region, but much higher atmospheric temperatures are needed to initiate the margin retreat.

We will add the description of these experiments to Section 2.4 and the figure R2.1 with its description to Section 3.2. This will further strengthen our conclusions that oceanic warming is responsible for the first bifurcation point, while atmospheric warming triggers the feedbacks responsible for the second bifurcation point.

Figure R2.1: a) Ice volume above flotation for three quasi-equilibrium simulations with a forcing rate of  $3\times10^{-5}$  K yr-1: (1) OCN-only simulation, maintaining regional summer temperature anomaly constant at -12 K (corresponding to the LGM-like state) during the warming branch and at +4 K (corresponding to the virtually ice-free state) during the cooling branch; (2) ATM-only simulation, maintaining ocean temperature anomaly constant at -4 K (corresponding to the LGM-like state) during the warming branch and at +1.3 K (corresponding to the virtually ice-free state) during the cooling branch; and (3) simulation with both atmospheric and oceanic forcing, same simulation presented in Fig. 3 of the manuscript. b)-d) The three snapshots of the OCN-only simulation marked with blue dots in a). e)-g) The three snapshots of the ATM-only simulation marked with red dots in a). The black contour lines in b)-g) indicate the surface elevation every 500 m starting from 0 and with a thicker line every 1000 m.

**Detailed comments**

L4: "global warming to 4K" would be changed to "threshold of ice-free state"?

Thanks for the suggestion, but we prefer to keep the original sentence, as we are referring to the forcing itself rather than the consequences of the forcing, which we believe is more accurate in this context.

L7-8: Please clarify that -12 K and +4K indicate regional summer temperature. Fixed.

L34-35: On the threshold of Greenland ice sheet, recent study (Gregory et al., 2020) addresses this topic, with the effect of ice sheet-climate interactions and the irreversibility of the Greenland ice sheet.

Thanks for the suggestion, we will add that reference to the introduction.

**L41: "regional summer atmospheric temperature" Where? Is it based on ice core site NGRIP?**

Yes, in central Greenland. There was an error in the reference. The data is from the merged product of Buizert et al. (2018), and we will clarify this in the manuscript.

L80: Is the extent of the ice shelf margin determined only by stress alone? Are there any geographical constraints like continental shelf break positions?

The extent of ice shelves is determined by ice dynamics and mass balance (surface mass balance, basal melting, and calving). However, since the presence of ice is considered implausible in the deep ocean, calving is applied where the bedrock depth exceeds 1000m. We will make it clear in the manuscript.

L83-84: As far as I understand, the REMBO needs specific humidity as the input. According to the model results' description, an increase in regional atmospheric air temperature leads to an increase in precipitation over Greenland (e.g., L266). How is the specific humidity treated in temperature changes? Is it assuming the relative humidity as the constant?

As developed in the answer to the first reviewer, the specific humidity Q at the boundary is given by:

$$Q = Q_{sat}(T) \cdot r$$
;

where  $Q_{sat}(T)$  is the saturation specific humidity, which depends on temperature following the Clausius-Clapeyron equation. Therefore, across the diagram, the temperature at the boundaries changes and the relative humidity is constant, but the specific humidity changes according to the temperature variations. In REMBO, the choice to maintain constant relative humidity rather than specific humidity is based on the fact that specific humidity has a stronger temperature dependence (Robinson et al., 2010).

We will clarify this in the manuscript.

L89: "100 km" which model's resolution? I suppose the resolution of ERA-40 is ~100 km; could you please clarify this?

REMBO resolution is 100km, we will clarify this in the manuscript.

L90: According to Robinson et al. (2010), the REMBO utilized empirical lapse rate feedback of 6.5 K/km for elevation correction. Is the same elevation-temperature feedback utilized in the current model used in these experiments?

Yes, it is the same value as in Robinson et al. (2010). We will clarify this in the manuscript.

L107: Is  $\delta P$  defined at every 16 km grid cell? It would be helpful to put the map of P and Pcorr in the supplemental Figure

No,  $\delta P$  is calculated in the REMBO model, which has a resolution of 100km; therefore,  $\delta P$  is calculated at every 100km grid cell, and it is smoothed. Thanks for the suggestion, we will clarify this in the manuscript and we will add the figure suggested to an appendix.

L109: What does "consistent field" mean?

We mean a field consistent with the boundary conditions at each moment (topography,  $\Delta T$ jja and specific humidity). Since the  $\delta P$  field can introduce a bias from present-day conditions, if it had no limits, it could introduce variations that would go beyond those related to the model's own bias. We will clarify this in the manuscript.

L128: Is The unit m/yr defined as "freshwater equivalent" mass balance? Or ice equivalent? Please clarify.

It is ice equivalent, we will clarify it in the manuscript.

L156: I think retaining insolation as present-day is one probable experimental design because summer insolation in the northern high latitude at the LGM is similar to present-day. However, I recommend the author consider adding one sensitivity experiment setting reduced sea level and setting LGM insolation values (used in the energy balance model REMBO as in equation 3) to assess the impact of these parameters.

This suggestion was also made by Reviewer 1. Therefore, we conducted sensitivity experiments examining the effects of varying insolation and sea-level conditions in the LGM-like initial state. As detailed in our response to Reviewer 1 (general comment #1), results show that under the insolation and sea level of the LGM some floating ice became grounded, but the overall ice-sheet configuration remains largely similar, particularly in the northeastern region. With LGM conditions (insolation and sea level), the first bifurcation point occurs at slightly higher temperatures, but still within the ocean forcing uncertainty range, and the overall system behavior is preserved.

L299: Please clarify at what degree of  $\Delta$ Tjja does basal melting activate? Clarified above.

L349-L354: In Bochow's (2023) experiment, oscillations were not observed in YELMO-REMBO. However, oscillations were observed in the experiment described in this article. I have identified that the experimental design is not identical with Bochow et al. (2023), the one is the scaling ratio of  $\Delta$ Tija and  $\Delta$ Tdjf (1.61 in Bochow et al. (2023) while the scaling ratio is 2 in this study. Are there other differences in how ocean melt is determined? I believe it would be beyond scope of this study to explain why oscillatory solutions appear in the experiments in the current setup. I recommend summarizing the differences in the experimental setup compared to Bochow et al. (2023) and stating that the existence of oscillatory solutions depends on the experimental setup.

Exactly, some of the differences with the experimental setup in Bochow et al. (2023) are: (1) the scaling ratio between summer and winter; (2) the melt parameters in the ocean melting equation ( $\kappa$  and  $B_{ref}$ ) and the scaling factor between ocean and atmosphere; and (3) the use of a bias correction in precipitation. However, the main difference lies in the experimental design itself.

In our experiments, oscillations occur under constant forcing conditions after a transient warming or cooling phase (Fig. B1 in the manuscript). Specifically, oscillations appear at +1.4 and +1.6 K, starting at approximately 200 kyr in the warming branch and 50 kyr in the cooling branch. In contrast, Bochow et al. (2023) applied a very rapid (nearly instantaneous) warming followed by rapid cooling to different convergence temperatures, which were then held constant for 100 kyr. They did not

explore what would occur if the equilibrium simulations were extended for longer periods (and at that exact temperature anomalies) and they didn't perform the cooling or regrowth branches. With longer timescales, the bedrock has more time to rebound isostatically, uplifting the surface to higher altitudes with lower temperatures. This favors ice regrowth and is the mechanism that, coupled with other feedbacks, we believe is responsible for the emergence of oscillations in our setup. However, since oscillations start on timescales longer than those explored by Bochow et al. (2023), direct comparison is not possible, as it is unclear whether extended simulations under their setup would exhibit similar behavior.

L373-L380: Please clarify that Honing (2023) defines temperature as the global mean temperature, which differs from this study and Robinson (2012).

Fixed.

L381-L386: Bochow et al. (2023) derives the relationship between global mean temperature and  $\Delta$ Tjja based on an analysis of the CMIP6 climate model historical and the SSP585 experiment. I recommend summarizing the method of relating global mean temperature and  $\Delta$ Tjja in the manuscript text.

We agree and we will clarify this methodology in the manuscript text. Following Bochow et al. (2023), we use the equation:

$$\Delta \mathrm{GMT_{PI}} = f imes \Delta T_{\mathrm{JJA}} + 0.5\,^{\circ}\mathrm{C}$$

where  $f = 1/1.19 \text{ K}^{-1}$ . This scaling factor represents the mean value derived from the CMIP6 SSP5-8.5 experiments (Extended Data Table 1 of Bochow et al., 2023). We use the factor derived from SSP5-8.5 rather than the one from historical simulations because in this case (L381-L386, talking about the second bifurcation point) our target regional temperature anomalies are above present-day values, making the SSP5-8.5 relationship more appropriate for future warming scenarios. Note that we use this scaling factor only when talking about the value of the second bifurcation point, this conversion wouldn't be appropriate for the first one.

**L373-L380**

Figure C1: It would be good to have a diagram with Mgl on the horizontal axis, which would allow us to consider whether the basal mass balance of the ocean or the surface mass balance of the atmosphere primarily determines hysteresis.

Given the new experiments performed (ATM-only and OCN-only) shown in Figure R2.1, which clearly demonstrate the effects of the ocean and atmosphere on stability, we believe it is not necessary to add the surface mass balance to this figure. Moreover, the purpose of this figure is to show the uncertainty in the first bifurcation point related to the basal melting scheme.

**Other minors:**

L5: Yelmo coupled with regional energy balance model REMBO It is mentioned in the next sentence, L10.

L47: "regional climate model REMBO" to "regional energy balance model REMBO" to make consistency.

Fixed.

L82: "regional climate model" to "regional energy balance model" Fixed.

**Figure A1: Could you please show the distribution of SMB in the current climate in this experimental setting, with a comparison to SMBMIP (Fettweis et al. 2020)?**

Yes, we think this is a great suggestion. However, while we simulate the present-day with the average conditions for the period 1958–2001, the ensemble from Fettweis et al. (2020) starts in 1980. This is why we also include outputs from MAR v3.14.3 at 10 km resolution, forced at its lateral boundaries by ERA5 and provided by Dr. Fettweis, which cover the period 1958–2001 and represent more recent simulations. Therefore, we believe it is more appropriate to include the latter in the revised version of the manuscript. We include the comparison below (Figure R2.2).

**Figure R2.2**: a) Yelmo-REMBO surface mass balance; b) GrSMBMIP ensemble mean for the period 1980-2001 (Frettweis et al. 2020); and c) MAR version 3.14.3 forced with ERA5, mean for the period 1958-2001. Values are expressed in meters of water equivalent per year.

**References:**

Alvarez-Solas, J., Banderas, R., Robinson, A., & Montoya, M. (2017). Oceanic forcing of the Eurasian Ice Sheet on millennial time scales during the Last Glacial Period. *Climate of the Past Discussions*, 2017, 1-19.

Albrecht, T., Winkelmann, R., & Levermann, A. (2020). Glacial-cycle simulations of the Antarctic Ice Sheet with the Parallel Ice Sheet Model (PISM)—Part 1: Boundary conditions and climatic forcing. *The Cryosphere*, *14*(2), 599-632.

Buizert, C., Keisling, B. A., Box, J. E., He, F., Carlson, A. E., Sinclair, G., & DeConto, R. M. (2018). Greenland-wide seasonal temperatures during the last deglaciation. *Geophysical Research Letters*, 45(4), 1905-1914.

Fettweis, X., Hofer, S., Krebs-Kanzow, U., Amory, C., Aoki, T., Berends, C. J., ... & Zolles, T. (2020). GrSMBMIP: intercomparison of the modelled 1980–2012 surface mass balance over the Greenland Ice Sheet. *The Cryosphere*, *14*(11), 3935-3958.

Golledge, N. R., Kowalewski, D. E., Naish, T. R., Levy, R. H., Fogwill, C. J., & Gasson, E. G. (2015). The multi-millennial Antarctic commitment to future sea-level rise. *Nature*, *526*(7573), 421-425.

Gregory, J. M., George, S. E., & Smith, R. S. (2020). Large and irreversible future decline of the Greenland ice sheet. *The Cryosphere*, *14*(12), 4299-4322.

Kusahara, K., Sato, T., Oka, A., Obase, T., Greve, R., Abe-Ouchi, A., & Hasumi, H. (2015). Modelling the Antarctic marine cryosphere at the Last Glacial Maximum. *Annals of Glaciology*, *56*(69), 425-435.

Obase, T., Abe-Ouchi, A., Kusahara, K., Hasumi, H., & Ohgaito, R. (2017). Responses of basal melting of Antarctic ice shelves to the climatic forcing of the Last Glacial Maximum and CO 2 doubling. *Journal of Climate*, *30*(10), 3473-3497.

Robinson, A., Calov, R., & Ganopolski, A. (2012). Multistability and critical thresholds of the Greenland ice sheet. *Nature Climate Change*, *2*(6), 429-432.

Robinson, A., Calov, R., & Ganopolski, A. (2010). An efficient regional energy-moisture balance model for simulation of the Greenland Ice Sheet response to climate change. *The Cryosphere*, 4(2), 129-144.

Tabone, I., Robinson, A., Alvarez-Solas, J., & Montoya, M. (2019a). Impact of millennial-scale oceanic variability on the Greenland ice-sheet evolution throughout the last glacial period. *Climate of the Past*, *15*(2), 593-609.

Tabone, I., Robinson, A., Alvarez-Solas, J., & Montoya, M. (2019b). Submarine melt as a potential trigger of the North East Greenland Ice Stream margin retreat during Marine Isotope Stage 3. *The Cryosphere*, *13*(7), 1911-1923.

Tabone, I., Robinson, A., Montoya, M., & Alvarez-Solas, J. (2024). Holocene thinning in central Greenland controlled by the Northeast Greenland Ice Stream. *Nature communications*, *15*(1), 6434.